# Towards Generalisable Time Series Understanding Across Domains

## Abstract

In natural language processing and computer vision, self-supervised pre-training on large datasets unlocks foundational model capabilities across domains and tasks. However, this potential has not yet been realised in time series analysis, where existing methods disregard the heterogeneous nature of time series characteristics. Time series are prevalent in many domains, including medicine, engineering, natural sciences, and finance, but their characteristics vary significantly in terms of variate count, inter-variate relationships, temporal dynamics, and sampling frequency. This inherent heterogeneity across domains prevents effective pre-training on large time series corpora. To address this issue, we introduce OTiS, an **o**pen model for general **ti**me **s**eries analysis, that has been specifically designed to handle multi-domain heterogeneity. We propose a novel pre-training paradigm including a tokeniser with learnable domain-specific signatures, a dual masking strategy to capture temporal causality, and a normalised cross-correlation loss to model long-range dependencies. Our model is pre-trained on a large corpus of $640,187$ samples and $11$ billion time points spanning $8$ distinct domains, enabling it to analyse time series from any (unseen) domain. In comprehensive experiments across $15$ diverse applications - including classification, regression, and forecasting - OTiS showcases its ability to accurately capture domain-specific data characteristics and demonstrates its competitiveness against state-of-the-art baselines. Our code and pre-trained weights are publicly available at `https://github.com/OTiS-official/OTiS`.

## 1 Introduction

In natural language processing (NLP) or computer vision (CV), generalisable language features, e.g. semantics and grammar (Radford et al., 2018; Touvron et al., 2023; Chowdhery et al., 2023), or visual features, e.g. edges and shapes (Geirhos et al., 2019; Dosovitskiy et al., 2021; Oquab et al., 2024), are learned from large-scale data. Self-supervised pre-training paradigms are designed to account for the specific properties of language (Radford et al., 2018; Touvron et al., 2023; Chowdhery et al., 2023) or imaging (Zhou et al., 2022; Cherti et al., 2023; Oquab et al., 2024), unlocking foundational model capabilities that apply to a wide range of domains and downstream tasks. This potential, however, remains largely unrealised in time series due to the lack of self-supervised pre-training paradigms that account for the heterogeneity of time series across domains.

Time series are widespread in everyday applications and play an important role in various domains, including medicine (Pirkis et al., 2021), engineering (Gasparin et al., 2022), natural sciences (Ravuri et al., 2021), and finance (Sezer et al., 2020). They differ substantially with respect to the number of variates, inter-variate relationships, temporal dynamics, and sampling frequency (Fawaz et al., 2018; Ismail Fawaz et al., 2019; Ye & Dai, 2021; Wickstrøm et al., 2022). For instance, standard 10-20 system electroencephalography (EEG) recordings come with up to $256$ variates (Jurcak et al., 2007), while most audio recordings have only $1$ (mono) or $2$ (stereo) variates. Weather data shows high periodicity, whereas financial data is exposed to long-term trends. Both domains encompass low-frequency data recorded on an hourly ($278\,\text{mHz}$), daily ($12\,\mu\text{Hz}$), or even monthly ($386\,\text{nHz}$) basis, while audio data is sampled at high frequencies of $44.1\,\text{kHz}$ or more. Overall, this heterogeneity across domains renders the extraction of generalisable time series features difficult (Fawaz et al., 2018; Gupta et al., 2020; Iwana & Uchida, 2021; Ye & Dai, 2021).

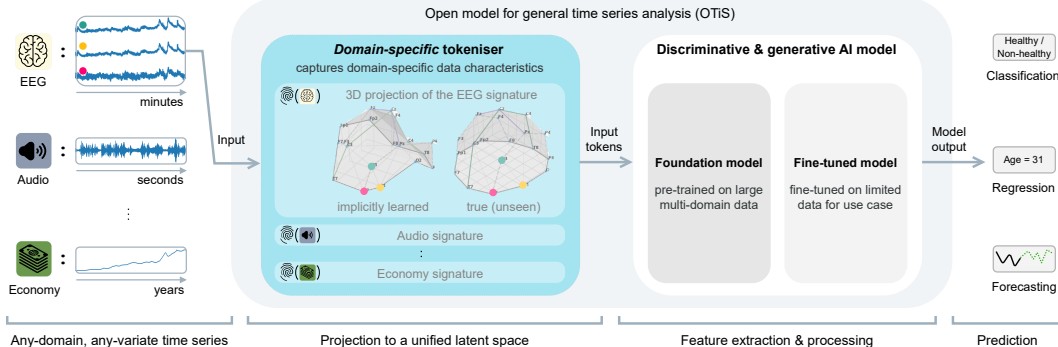

Figure 1: Overview of `OTiS`. Pre-trained on a large corpus of time series from diverse domains, `OTiS` enables general time series analysis. Its domain-specific tokeniser addresses time series heterogeneity across domains - including different numbers of variates, inter-variate relationships, temporal dynamics, and sampling frequencies - by learning unique domain signatures. After pre-training, the model can be fine-tuned on limited data from any domain, including previously unseen ones, to perform various tasks such as classification, regression, and forecasting.

While most existing self-supervised pre-training methods for time series are limited to single domains (Wu et al., 2021; 2022a; Nie et al., 2023; Dong et al., 2024; Jiang et al., 2024), recent works propose simple techniques to incorporate time series from multiple domains (Yang et al., 2024; Das et al., 2024; Woo et al., 2024; Liu et al., 2024). These works for instance crop all time series into segments of unified size (Jiang et al., 2024), resample them to a uniform frequency (Yang et al., 2024), or analyse each variate of a multi-variate time series independently (Liu et al., 2024). While these naive techniques address differences in sampling frequency and variate count, they degrade the original time series and neglect the critical inter-variate relationships and temporal dynamics required for effective real-world analysis. Consequently, there is a clear need for pre-training strategies that adequately handle heterogeneity in time series to unlock foundational model capabilities.

In this work, we propose a novel multi-domain pre-training paradigm that addresses the full spectrum of time series heterogeneity across domains. Our approach facilitates the comprehensive extraction of generalisable features from diverse time series. Pre-trained on a large corpus of publicly available data, our **o**pen model for general **ti**me **s**eries analysis (`OTiS`) can be fine-tuned on limited data of any (unseen) domain to perform a variety of downstream tasks, as showcased in Figure 1.

Our key contributions can be summarised as follows:

1. We present `OTiS`, an **o**pen model for general **ti**me **s**eries analysis, with our entire pipeline and pre-trained weights publicly available at https://github.com/OTiS-official/OTiS.

2. We propose a novel pre-training paradigm based on masked data modelling to address heterogeneity in multi-domain time series. Our approach includes a novel tokeniser with learnable signatures to capture domain-specific data characteristics, a dual masking strategy to learn temporal causality, and a normalised cross-correlation loss to model long-range dependencies.

3. We pre-train `OTiS` on a large corpus of publicly available time series from 8 domains, spanning medicine, engineering, natural sciences, and finance. With $640,187$ samples and 11 billion time points, this corpus represents diverse time series characteristics, enabling generalisable feature extraction.

4. We evaluate `OTiS` across 15 downstream applications, including classification, regression, and forecasting. Our comprehensive analysis demonstrates that `OTiS` accurately captures domain-specific data characteristics and is competitive with both specialised and general state-of-the-art (SOTA) models, achieving new SOTA performance in 10 tasks. Notably, none of the baselines is capable of performing all the tasks covered by `OTiS`.

## 2 RELATED WORKS

### 2.1 SELF-SUPERVISED LEARNING FOR TIME SERIES

Time series vary significantly across domains, with differences in the number of variates, inter-variate relationships, temporal dynamics, and sampling frequencies. Due to this inherent heterogeneity, most existing works focus on pre-training models within a single domain (Oreshkin et al., 2019; Tang et al., 2020; Wu et al., 2021; Zhou et al., 2021; Wu et al., 2022a; Woo et al., 2022; Yue et al., 2022; Zhang et al., 2022; Li et al., 2023; Nie et al., 2023; Zeng et al., 2023; Dong et al., 2024). To develop more generalisable time series models, recent methods have explored multi-domain pre-training by addressing certain aspects of the heterogeneity, such as differences in variate count and sampling frequency. For instance, Liu et al. (2024) treat each variate in multi-variate time series independently to standardise generative tasks like forecasting, while Goswami et al. (2024) extend uni-variate analysis to discriminative tasks like classification. Similarly, Jiang et al. (2024) and Yang et al. (2024) standardise time series by cropping them into segments of predefined size and resampling them to a uniform frequency, respectively, to enable general classification capabilities in medical domains.

While partially addressing time series heterogeneity, these methods limit model capabilities for general time series analysis. Standardisation techniques like cropping or resampling may distort inter-variate relationships, temporal dynamics, and long-range dependencies. Additionally, many of these approaches are tailored to specific applications, such as generative tasks (Das et al., 2024; Liu et al., 2024; Woo et al., 2024), or focused on particular domains like medicine (Jiang et al., 2024; Yang et al., 2024). Moreover, recent foundational models (Das et al., 2024; Goswami et al., 2024; Liu et al., 2024) focus on uni-variate analysis, ignoring crucial inter-variate relationships essential for real-world applications, such as disease prediction (Schoffelen & Gross, 2009; Wu et al., 2022b). Our study aims to overcome these limitations by fully addressing heterogeneity of multi-domain time series, establishing a foundation for general time series analysis across domains and tasks.

### 2.2 TIME SERIES TOKENISATION

Transformers (Vaswani et al., 2017) have emerged as the preferred architecture for foundational models in NLP and CV due to their scalability (Kaplan et al., 2020; Gordon et al., 2021; Alabdulmohsin et al., 2022), enabling the training of models in the magnitude of 100 billion parameters (Chowdhery et al., 2023; Touvron et al., 2023; Oquab et al., 2024; Ravi et al., 2024). To utilise a Transformer for time series analysis, a tokeniser is required to map the time series into a compact latent space. Current methods (Jin et al., 2023; Nie et al., 2023; Zhou et al., 2023; Das et al., 2024; Goswami et al., 2024; Jiang et al., 2024; Liu et al., 2024; Woo et al., 2024; Yang et al., 2024) follow established techniques from NLP and CV, dividing time series into patches of pre-defined size. These patches are then flattened into a 1D sequence, with positional embeddings used to retain positional information. While uni-variate models (Nie et al., 2023; Das et al., 2024; Goswami et al., 2024; Liu et al., 2024) consider only temporal positions, multi-variate approaches (Woo et al., 2024; Yang et al., 2024; Jiang et al., 2024) account for both temporal and variate positions. However, none of these methods address the unique characteristics of variates, mistakenly assuming that the relationships between variates are identical across domains. Our work seeks to adapt the tokenisation process to preserve the domain-specific relationships between variates.

## 3 METHODS

In this work, we present a novel multi-domain pre-training paradigm that enables generalisable feature extraction from large, heterogeneous time series corpora. We introduce a domain-specific tokeniser with learnable signatures to address heterogeneity in multi-domain time series, as described in Section 3.1. We tailor masked data modelling (MDM) for multi-domain time series to pre-train our **o**pen model for general **ti**me **s**eries analysis (OTiS) on a large, heterogeneous corpus, as detailed in Section 3.2. In particular, we propose normalised cross-correlation as a loss term to capture global temporal dynamics in time series, as explained in Section 3.3. Moreover, we introduce a dual masking strategy to capture bidirectional relationships and temporal causality, essential for general time series analysis, as described in Section 3.4. After pre-training, we fine-tune OTiS on limited data to perform a variety of downstream tasks in any - including previously unseen - domain, as outlined in Section 3.5. A graphical visualisation of our method is provided in Figure 2.

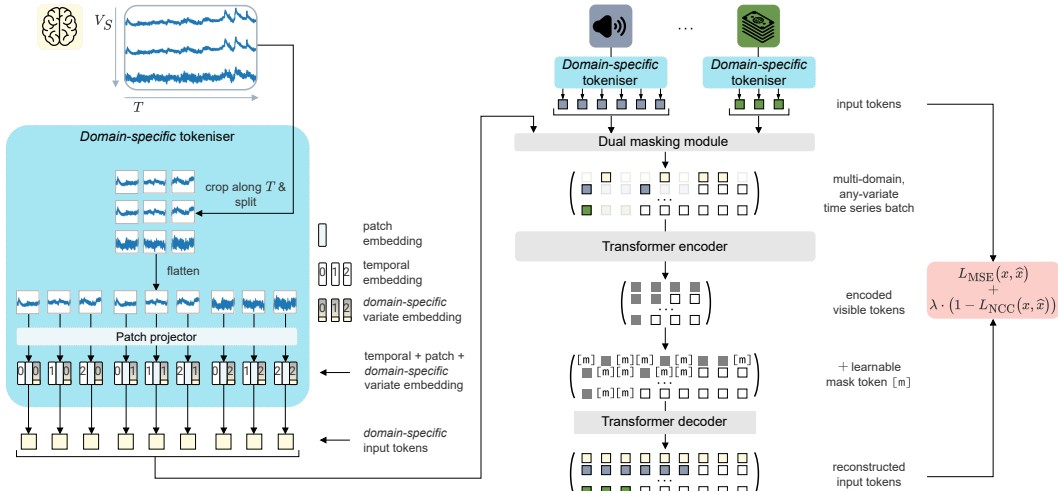

Figure 2: Architecture of OTiS. During pre-training, batches of time series from diverse domains are processed using a domain-specific tokeniser. This tokeniser splits a time series into fixed-size patches, which are then embedded using a patch projector shared across all variates and domains. A temporal embedding and a domain-specific variate embedding are added to each patch embedding. A dual masking strategy is employed to mask the resulting input tokens. The reconstruction of the multi-domain input tokens is guided using an auxiliary normalised cross-correlation (NCC) loss.

## 3.1 DOMAIN-SPECIFIC TOKENISER

**Overview.** Assume a time series sample $\boldsymbol{X} \in \mathbb{R}^{V_S \times T}$ from domain $S$, where $V_S$ denotes the number of variates specific to $S$ and $T$ denotes the number of time points. We randomly crop or zero-pad $\boldsymbol{X}$ to a fixed context length of $\overline{T}$ time points. We then split it into $T'$ temporal patches of size $P$ along the time dimension, resulting in $V_S \cdot T'$ patches $\boldsymbol{x}_{v,t} \in \mathbb{R}^{1 \times P}$, where $v \in \{1, \ldots, V_S\}$ and $t \in \{1, \ldots, T'\}$.

Next, we embed these patches using a shared patch projector across all variates and domains, resulting in patch embeddings $e^{\mathcal{P}}(\boldsymbol{x}_{v,t}) = \boldsymbol{e}_{v,t}^{\mathcal{P}} \in \mathbb{R}^{1 \times D}$, where $D$ denotes the model dimension. The patch projector consists of a 1D convolutional layer followed by layer normalisation and GELU activation.

The permutation-equivariant nature of Transformers (Vaswani et al., 2017) requires the use of positional embeddings to accurately capture the inherent relationships in the input data. Initially introduced for 1D textual token sequences (Vaswani et al., 2017), positional embeddings simply introduce an ordering into the input sequence. Modern implementations further extend their capabilities to encode more complex geometric information, such as 2D spatial (Dosovitskiy et al., 2021) or graph (Kreuzer et al., 2021) structures. For the analysis of any-variate time series, we distinguish between the temporal and variate structure. The temporal structure is equivalent to a sequential 1D structure, such that we use standard 1D sinusoidal embeddings $e^{\mathcal{T}}(\boldsymbol{x}_{v,t}) = \boldsymbol{e}_t^{\mathcal{T}} \in \mathbb{R}^{1 \times D}$.

The variate structure exhibits great heterogeneity across domains. In domains with uni-variate and two-variate data, such as mono and stereo audio, the structure is either trivial or only requires a basic distinction between variates. In other domains, however, the variate structure may represent more complex relationships, such as 3D manifolds for electroencephalography (EEG) or electrocardiography (ECG) data, or be of non-spatial nature, such as for financial data. Hence, we introduce learnable *domain-specific* variate embeddings to adequately address the heterogeneity across domains. These embeddings, denoted as $e_S^{\mathcal{V}}(\boldsymbol{x}_{v,t}) = \boldsymbol{e}_{S,v}^{\mathcal{V}} \in \mathbb{R}^{1 \times D}$ for each variate $v$ in domain $S$, are designed to model the unique properties of a domain. They capture the inter-variate relationships and temporal dynamics specific to domain $S$, forming what can be considered as the *signature* of the very domain.

Finally, the patch, temporal, and domain-specific variate embeddings are summed to form the input token $\boldsymbol{e}_{v,t} = \boldsymbol{e}_{v,t}^{\mathcal{P}} + \boldsymbol{e}_t^{\mathcal{T}} + \boldsymbol{e}_{S,v}^{\mathcal{V}} \in \mathbb{R}^{1 \times D}$. These input tokens collectively constitute the final input sequence $\boldsymbol{E} \in \mathbb{R}^{(V_S \cdot T') \times D}$. To support batches of any-variate time series from multiple domains, we

pad the variate dimension to the maximum number of variates in a batch $\overline{V} = \max_S V_S$. For samples where $V_S < \overline{V}$ or $T < \overline{T}$, attention masking is used to ensure that padded variate or temporal tokens are ignored. The domain-specific tokeniser is trained end-to-end with the Transformer layers.

**Definition of (Sub-)Domains.** The domain-specific tokeniser is designed to integrate different datasets within a domain. Consider two EEG datasets, TDBrain (Van Dijk et al., 2022) and SEED (Zheng & Lu, 2015), which share 19 identical variates but have different sampling frequencies of $500\,\mathrm{Hz}$ and $200\,\mathrm{Hz}$, respectively. In this case, a single EEG-specific tokeniser ($V_{\mathrm{EEG}} = 19$) is sufficient to accommodate both sampling frequencies, i.e. $\boldsymbol{E}^{\mathcal{V}}_{\mathrm{EEG\text{-}TDBrain}} = \boldsymbol{E}^{\mathcal{V}}_{\mathrm{EEG\text{-}SEED}} = [\boldsymbol{e}^{\mathcal{V}}_{\mathrm{EEG},1}, \ldots, \boldsymbol{e}^{\mathcal{V}}_{\mathrm{EEG},19}]^{\top} \in \mathbb{R}^{19 \times D}$, as demonstrated in our experiments in Section 4. Note that while these positional embeddings are agnostic to variate ordering, we simplify processing by aligning the variate order across datasets within the same domain. Consider another EEG dataset, LEMON (Babayan et al., 2019), which includes 62 electrodes. Of these, 15 overlap with the electrodes in TDBrain (Van Dijk et al., 2022) and SEED (Zheng & Lu, 2015), while the remaining 47 are unique to LEMON (Babayan et al., 2019). In this scenario, the EEG-specific tokeniser can be extended by the 47 new variates ($V_{\mathrm{EEG}} = 66$), such that $\boldsymbol{E}^{\mathcal{V}}_{\mathrm{EEG\text{-}LEMON}} = [\boldsymbol{e}^{\mathcal{V}}_{\mathrm{EEG},1}, \ldots, \boldsymbol{e}^{\mathcal{V}}_{\mathrm{EEG},15}, \boldsymbol{e}^{\mathcal{V}}_{\mathrm{EEG},20}, \ldots, \boldsymbol{e}^{\mathcal{V}}_{\mathrm{EEG},66}]^{\top} \in \mathbb{R}^{62 \times D}$. In this way, different datasets can be combined to approximate the underlying data distribution of a domain $S$, e.g. EEG, enabling the creation of large and diverse time series corpora.

**Multi-Variate or Uni-Variate Analysis?** Consider the Electricity dataset (UCI, 2024), which contains electricity consumption data for 321 households recorded from 2012 to 2014. These 321 observations are sampled from an underlying population and are assumed to be independent and identically distributed (*i.i.d.*). In this scenario, we perform a uni-variate analysis ($V_{\mathrm{Electricity}} = 1$) of the data, initialising a single Electricity-specific variate embedding that models the hourly consumption of a household. In contrast, the Weather dataset (Wetterstation, 2024) contains 21 climatological indicators, such as air temperature, precipitation, and wind speed, which are not *i.i.d.* because they directly interact and correlate with one another. Therefore, a multi-variate analysis ($V_{\mathrm{Weather}} = 21$) is conducted to account for the dependencies and interactions between the observations.

## 3.2 PRE-TRAINING ON MULTI-DOMAIN TIME SERIES

We pre-train our model using masked data modelling (MDM) (He et al., 2022) to learn generalisable time series features across domains. We mask a subset of input tokens and only encode the visible (i.e. non-masked) tokens using an encoder $f(\cdot)$. Afterwards, we complement the encoded tokens with learnable mask tokens and feed them to a decoder $g(\cdot)$, reconstructing the original input tokens.

More precisely, we draw a binary mask $\boldsymbol{m} \in \{0,1\}^{V_S \cdot T'}$, following the dual masking strategy proposed in Section 3.4, and apply it to the input sequence $\boldsymbol{E} \in \mathbb{R}^{(V_S \cdot T') \times D}$. Thus, we obtain a visible view $\boldsymbol{E}[\boldsymbol{m}] \in \mathbb{R}^{N_1 \times D}$, where $N_1 = \sum_{v=1}^{V_S} \sum_{t=1}^{T'} m_{v,t}$ and $N_0 = (V_S \cdot T') - N_1$ denote the number of visible and masked tokens, respectively. The visible view $\boldsymbol{E}[\boldsymbol{m}]$ is then fed to the encoder $f(\cdot)$ to compute the token features $\boldsymbol{H} \in \mathbb{R}^{N_1 \times D}$:

$$\boldsymbol{H} = f(\boldsymbol{E}[\boldsymbol{m}]). \tag{1}$$

To reconstruct the original input, these token features are fed to the decoder $g(\cdot)$ together with a special, learnable mask token $\boldsymbol{e}^{\mathcal{M}} \in \mathbb{R}^{1 \times D}$, that is inserted at the masked positions where $m_{v,t} = 0$:

$$\boldsymbol{h}'_{v,t} = \begin{cases} \boldsymbol{h}_{v,t} & \text{if } m_{v,t} = 1 \\ \boldsymbol{e}^{\mathcal{M}} & \text{if } m_{v,t} = 0 \end{cases}, \tag{2}$$

such that $\boldsymbol{H}' \in \mathbb{R}^{(V_S \cdot T') \times D}$. The decoder $g(\cdot)$ then predicts the reconstructed input $\widehat{\boldsymbol{X}} \in \mathbb{R}^{V_S \times (T' \cdot P)}$:

$$\widehat{\boldsymbol{X}} = g(\boldsymbol{H}'), \tag{3}$$

where $(T' \cdot P) = \overline{T}$, i.e. the context length specified in time points. Eventually, the domain-specific tokeniser described in Section 3.1, the encoder $f(\cdot)$, and the decoder $g(\cdot)$ are optimised end-to-end using the mean squared error (MSE) loss on all reconstructed input tokens:

$$\mathcal{L}_{\mathrm{MSE}} = \frac{1}{V_S \cdot T'} \sum_{v=1}^{V_S} \sum_{t=1}^{T'} \|\boldsymbol{x}_{v,t} - \widehat{\boldsymbol{x}}_{v,t}\|_2^2. \tag{4}$$

### 3.3 NORMALISED CROSS-CORRELATION LOSS

MDM focuses on reconstructing masked parts of the data, emphasising *local* patterns through the MSE loss (4). However, time series often exhibit long-range dependencies, where past values influence future outcomes over extended periods. To accurately capture these *global* patterns, we introduce normalised cross-correlation (NCC) as a loss term in MDM for time series:

$$\mathcal{L}_{\text{NCC}} = \frac{1}{V_S \cdot \overline{T}} \sum_{v=1}^{V_S} \sum_{t=1}^{\overline{T}} \frac{1}{\sigma_{\boldsymbol{x}_v} \sigma_{\widehat{\boldsymbol{x}}_v}} (x_{v,t} - \mu_{\boldsymbol{x}_v})(\widehat{x}_{v,t} - \mu_{\widehat{\boldsymbol{x}}_v}) \in [-1, 1], \tag{5}$$

where $\mu$ and $\sigma$ denote the mean and standard deviation, respectively. Hence, to capture both local and global temporal dynamics, the total loss used to optimise OTiS is defined as

$$\mathcal{L} = \mathcal{L}_{\text{MSE}} + \lambda \cdot (1 - \mathcal{L}_{\text{NCC}}), \tag{6}$$

where $\lambda$ is empirically set to $0.1$ during pre-training.

### 3.4 DUAL MASKING STRATEGY

We design the masking strategy to enhance foundational model capabilites in time series analysis. Specifically, we randomly select between two masking schemes during pre-training, namely random masking and post-fix masking. In 75 % of cases, we apply random masking, where each $m_{v,t}$ is independently sampled from a Bernoulli distribution with probability $p = 1 - \rho$, with $\rho$ denoting the masking ratio (i.e. $m_{v,t} \sim \text{Bernoulli}(1 - \rho)$). This encourages the model to learn complex inter-variate relationships across the entire time series. In the remaining 25 % of cases, we employ post-fix masking, which masks the second half of the temporal dimension, leaving only the first half visible (i.e. $m_{v,t} = \mathbb{1}_{[t \leq T'/2]}$). The prediction of future values solely based on past observations simulates real-world forecasting conditions, helping the model to capture temporal causality. Overall, this dual masking strategy enables OTiS to learn both bidirectional relationships and temporal causality, which are essential for general time series analysis.

### 3.5 FINE-TUNING & INFERENCE ON (UNSEEN) TARGET DOMAINS

**Inclusion of Unseen Domains.** For a new domain $S$, a randomly initialised variate embedding $\boldsymbol{E}_S^{\mathcal{V}} \in \mathbb{R}^{V_S \times D}$ is introduced. The domain-specific tokeniser is then fine-tuned alongside the encoder $f(\cdot)$, and, if required, the decoder $g(\cdot)$, for the specific downstream task, as described in the following.

**Classification & Regression.** We use the encoder $f(\cdot)$ and the unmasked input sequence $\boldsymbol{E}$ to compute all token features $\boldsymbol{H} = f(\boldsymbol{E}) \in \mathbb{R}^{(V_S \cdot T') \times D}$. We average-pool these features into a global token $\boldsymbol{h}^* \in \mathbb{R}^{1 \times D}$, which we feed through a linear layer to obtain the final model prediction. We optimise a cross-entropy and MSE loss for the classification and regression tasks, respectively.

**Forecasting.** We apply post-fix masking to generate a binary mask $\boldsymbol{m} \in \{0, 1\}^{V_S \cdot T'}$ for the forecasting task. The encoder $f(\cdot)$ is used to compute the visible token features $\boldsymbol{H} \in \mathbb{R}^{N_1 \times D}$. We then concatenate the sequence with learnable mask tokens to form $\boldsymbol{H}' \in \mathbb{R}^{(V_S \cdot T') \times D}$, which is passed through the decoder $g(\cdot)$ to produce the final output. We optimise the MSE loss together with the NCC loss term over all reconstructed input tokens.

## 4 EXPERIMENTS & RESULTS

### 4.1 MODEL VARIANTS AND IMPLEMENTATION DETAILS

We introduce OTiS in three different configurations, Base, Large, and Huge, with their specific architectures described in Appendix C.1, to explore scaling laws with respect to the model size. We set the patch size and stride to $P = 24$, respectively, to split the time series into $T' = \frac{\overline{T}}{P}$ non-overlapping patches along the time dimension. For pre-training, the context length specified in time points is set to $\overline{T} = 1008$, resulting in $T' = 42$ sinusoidal temporal embeddings. If longer context lengths are

Table 1: Overview of our large and diverse pre-training corpus. The corpus is built with unlabelled data from eight domains, encompassing medicine, engineering, natural sciences, and finance.

| Domain $S$ | Name | Samples | Variates $V_S$ | Time points | Frequency | Disk size |
|---|---|---|---|---|---|---|
| ECG | MIMIC-IV-ECG 2023 | $400,000$ | 12 | $5,000$ | 500 Hz | 90 GB |
| Temperature | DWD 2024 | $203,340$ | 1 | 720 | (hourly) $278\,\mu$Hz | 614 MB |
| Audio (stereo) | AudioSet-20K 2017 | $16,123$ | 2 | $441,000$ | 44.1 kHz | 53 GB |
| Audio (mono) | AudioSet-20K 2017 | $3,491$ | 1 | $441,000$ | 44.1 kHz | 6 GB |
| Electromechanics | FD-A 2016 | $13,640$ | 1 | $5,120$ | 64 kHz | 161 MB |
| EEG | TDBrain 2022 | $2,692$ | 19 | $60,000$ | 500 Hz | 12 GB |
| EEG | SEED 2015 | 675 | 19 | $37,000$ | 200 Hz | 2 GB |
| Banking | NN5 2012 | 111 | 1 | 971 | (daily) $12\,\mu$Hz | 370 KB |
| Economics | FRED-MD 2016 | 107 | 1 | 728 | (monthly) 386 nHz | 330 KB |
| Economics | Exchange 2018 | 8 | 1 | $7,588$ | (daily) $12\,\mu$Hz | 240 KB |
| | | **640,187** | | **11,052,756,981** | | **164 GB** |

required during fine-tuning, these embeddings are linearly interpolated (i.e. $T'_{\text{ft}} \geq 42$) to offer greater flexibility for downstream applications. We tune the hyperparameters for pre-training and fine-tuning as described in Appendix C. An overview of the computational costs is provided in Appendix D.

## 4.2 LARGE AND DIVERSE PRE-TRAINING CORPUS

We aim to develop a general time series model that fully handles the heterogeneity in real-world data. Specifically, our model is designed to handle time series with different variate counts $V_S$, inter-variate relationships, temporal dynamics, and sampling frequency, ensuring flexibility for downstream tasks. To this end, we pre-train our model on a large and diverse corpus of publicly available data spanning 8 domains, with a total of $640,187$ samples and 11 billion time points, as summarised in Table 1. A detailed description of the datasets included in our pre-training corpus can be found in Appendix A. The time series corpus is split into $612,394$ training and $27,793$ validation samples for pre-training.

## 4.3 BENCHMARKING ACROSS DOMAINS AND TASKS

To evaluate OTiS in real-world settings, we conduct experiments on three key use cases in time series analysis: classification, regression, and forecasting. We use 10 datasets across 8 domains to compare our model against 21 specialised and general baselines as outlined in Appendix B. The baselines include 11 target-specific models (either fully supervised or pre-trained and fine-tuned on target data), 6 general models (pre-trained on external data and fine-tuned on target data), and 4 foundation models (pre-trained on large corpora and fine-tuned on target data). We follow established data splitting and evaluation procedures for classification (Zhang et al., 2022), regression (Turgut et al., 2023), and forecasting (Zhou et al., 2021), with results reported across five seeds set during fine-tuning.

The experiments reveal that OTiS is a powerful feature extractor for time series analysis, achieving state-of-the-art performance on 10 out of 15 diverse benchmarks. The classification results in Table 2a highlight its particular strength in processing long time series, as indicated by a huge performance boost on FD-B ($\overline{T}_{\text{FD-B}} = 5112$). We also find that pre-training across domains is more effective than domain-specific pre-training. For instance, in the regression tasks shown in Table 2b, OTiS excels at predicting cardiac phenotypes, outperforming baselines pre-trained solely on ECG data (MAE) and even multimodally pre-trained baselines (CM-AE and MMCL). These results stress the strength of pre-training across domains for generalisable feature extraction, enabling OTiS to achieve superior performance even in unseen domains, as shown by the forecasting results in Table 3 and Appendix H. Zero-shot and linear probing experiments detailed in Appendix F further demonstrate OTiS' generalisability, resulting in competitive performance on Epilepsy, LVESV, and Weather prediction.

## 4.4 DOMAIN SIGNATURE ANALYSIS

A key component of OTiS is its use of domain-specific variate embeddings. While these embeddings are randomly initialised, we expect them to capture unique domain characteristics during training,

Table 2: Classification and regression performance on a total of 9 benchmark tasks. OTiS is competitive with specialised baselines, setting new state-of-the-art on 6 tasks and even outperforming the multimodal CM-AE and MMCL. This demonstrates the capability of OTiS to extract high-level semantics. Best score in **bold**, second best underlined. • indicates tasks in previously unseen domains.

(a) Classification [Accuracy (ACC ↑) in %]

| Model | Epilepsy | FD-B | Gesture• | EMG• |
|---|---|---|---|---|
| SimCLR 2020 | 90.71 | 49.17 | 48.04 | 61.46 |
| TimesNet 2022a | 94.01 | 56.86 | 59.79 | 91.22 |
| CoST 2022 | 88.40 | 47.06 | 68.33 | 53.65 |
| TS2Vec 2022 | 93.95 | 47.90 | 69.17 | 78.54 |
| TF-C 2022 | 94.95 | 69.38 | 76.42 | 81.71 |
| Ti-MAE 2023 | 89.71 | 60.88 | 71.88 | 69.99 |
| SimMTM 2024 | **95.49** | 69.40 | **80.00** | 97.56 |
| OTiS-Base | 94.25 | **99.24** | 63.61 | 97.56 |
| OTiS-Large | 94.03 | 98.62 | 62.50 | 98.37 |
| OTiS-Huge | 91.48 | 98.32 | 63.61 | **98.37** |
| OTiS₀° | 95.18 | 61.32 | 51.67 | 95.12 |

° Zero-shot predictions of OTiS-Base.

(b) Regression [R-squared ($R^2$ ↑)]

| Model | LVEDV | LVESV | LVSV | LVEF | LVM |
|---|---|---|---|---|---|
| iTransf. 2023 | 0.307 | 0.279 | 0.227 | 0.070 | 0.361 |
| ViT 2023 | 0.409 | 0.396 | 0.299 | 0.175 | 0.469 |
| MAE 2023 | 0.486 | 0.482 | 0.359 | 0.237 | 0.573 |
| CM-AE* 2023 | 0.451 | 0.380 | 0.316 | 0.103 | 0.536 |
| MMCL* 2023 | 0.504 | 0.503 | 0.370 | 0.250 | **0.608** |
| OTiS-Base | **0.509** | 0.512 | **0.391** | **0.292** | 0.592 |
| OTiS-Large | 0.504 | 0.503 | 0.371 | 0.267 | 0.592 |
| OTiS-Huge | 0.505 | **0.510** | 0.376 | 0.281 | 0.593 |
| OTiS$_{LP}$° | 0.414 | 0.394 | 0.279 | 0.161 | 0.453 |

* Models incorporate paired imaging data during pre-training.

° Linear probing of OTiS-Base.

Table 3: Forecasting performance on 6 benchmark tasks. OTiS is competitive with specialised and general baselines, setting new state-of-the-art on 4 tasks and showcasing its ability to capture local time series features. A forecasting horizon of 96 time points is predicted from the past 336 (*512, [+]904) time points. Mean squared error (MSE ↓) is reported. Best score in **bold**, second best underlined. • indicates tasks in previously unseen domains.

| Model | ETTh1• | ETTh2• | ETTm1• | ETTm2• | Weather• | Electricity• |
|---|---|---|---|---|---|---|
| N-BEATS 2019 | 0.399 | 0.327 | 0.318 | 0.197 | 0.152 | 0.131 |
| Autoformer 2021 | 0.435 | 0.332 | 0.510 | 0.205 | 0.249 | 0.196 |
| TimesNet 2022a | 0.384 | 0.340 | 0.338 | 0.187 | 0.172 | 0.168 |
| DLinear 2023 | 0.375 | 0.289 | 0.299 | 0.167 | 0.176 | 0.140 |
| PatchTST 2023 | **0.370** | 0.274 | 0.293 | 0.166 | 0.149 | 0.129 |
| Time-LLM[‡] 2023 | 0.408 | 0.286 | 0.384 | 0.181 | [†] | [†] |
| GPT4TS 2023 | 0.376 | 0.285 | **0.292** | 0.173 | 0.162 | 0.139 |
| MOMENT* 2024 | 0.387 | 0.288 | 0.293 | 0.170 | 0.154 | 0.136 |
| MOIRAI[+] 2024 | 0.375 | 0.277 | 0.335 | 0.189 | 0.167 | 0.152 |
| OTiS-Base | 0.424 | 0.212 | 0.337 | **0.161** | **0.139** | 0.128 |
| OTiS-Large | 0.446 | **0.205** | 0.362 | 0.173 | 0.142 | **0.127** |
| OTiS-Huge | 0.461 | 0.215 | 0.384 | 0.181 | 0.149 | 0.132 |
| OTiS$_{VE}$° | 0.434 | 0.217 | 0.396 | 0.182 | 0.149 | 0.164 |

[†] Experiments could not be conducted on a single NVIDIA RTX A6000-48GB GPU.

[‡] Model incorporates paired text data during pre-training and fine-tuning.

° Predictions of OTiS-Base with only the domain-specific variate embeddings (VE) and mask token trained.

eventually serving as the signature of their respective domain. To validate this hypothesis, we analyse the domain-specific variate embeddings after pre-training using principal component analysis (PCA).

First, we find that OTiS unifies time series from diverse domains into a meaningful latent space, where embeddings of domains with shared high-level semantics cluster together, as depicted in Appendix E.1. For example, embeddings of mono and stereo audio group closely, as do those of banking and economics. Moreover, EEG-specific embeddings are clearly separated and ECG-specific embeddings form a tight cluster.

Second, we observe that OTiS preserves the low-level semantics of a domain, such as the relationships between variates. To explore this, we focus on EEG, where variates correspond to electrodes with defined spatial positions (either in 3D space or 2D on the scalp), making it an ideal domain for studying

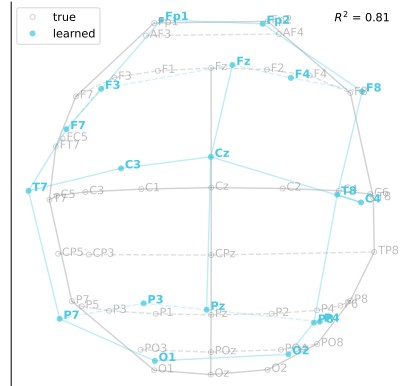 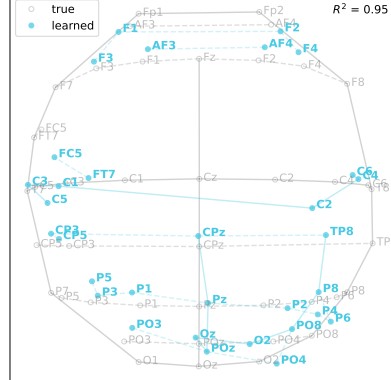

Figure 3: First two principal components of the EEG-specific variate embeddings, overlaid on the true EEG electrode layout. (**Left**) Embeddings of 10-20 system EEG recordings with 19 electrodes learned during pre-training. (**Right**) Embeddings of previously unseen EEG recordings with 32 electrodes learned during fine-tuning. The embeddings accurately reflect the spatial electrode layout, as confirmed by high correlations ($R^2$) between the PCA projections ● and the true layout ○.

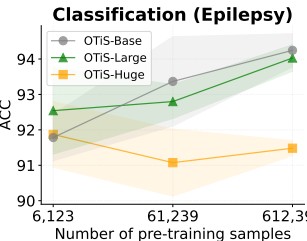 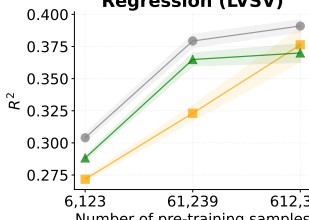 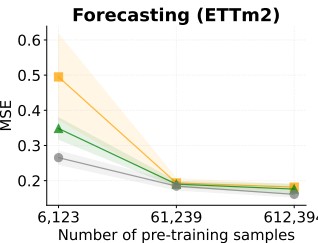

Figure 4: Performance of `OTiS` with different numbers of pre-training samples. Shaded regions indicate the standard deviation across 5 seeds. Increasing dataset size generally improves downstream performance. Scaling model size requires even larger pre-training corpora to be effective.

inter-variate relationships. Our analysis includes (i) variate embeddings of 10-20 system EEG recordings with 19 electrodes learned during multi-domain pre-training, and (ii) variate embeddings of previously unseen EEG recordings with 32 electrodes learned during fine-tuning. We determine the first three principal components of the learned EEG-specific variate embeddings (visualised in Appendix E.2.1) and find that they explain (i) $74.7\,\%$ and (ii) $87.9\,\%$ of the variance. These findings suggest that the embeddings reflect the true EEG electrode layout. To approve this hypothesis, we linearly align the 3D PCA projections with the true 3D electrode coordinates and quantify their correlation, as detailed in Appendix E.2.1. We observe $R^2$ values of (i) 0.81 and (ii) 0.95, confirming that the learned variate embeddings accurately capture the true electrode layout, as visualised in Figure 3. Further analyses of ECG- and Weather-specific variate embeddings, presented in Appendix E.2.2, strengthen `OTiS`' ability to model complex inter-variate relationships across diverse domains.

## 4.5 SCALING STUDY

We analyse the scaling behaviour of `OTiS` with respect to model and dataset size. To this end, we subsample the pre-training data to $10\,\%$ and $1\,\%$ of its original size, ensuring that each subset is fully contained within the corresponding superset. We evaluate the downstream performance of all `OTiS` variants across classification, regression, and forecasting tasks, as depicted in Figure 4.

The experiments demonstrate that downstream performance generally scales with dataset size, achieving the best results with the full pre-training dataset. This trend, however, does not directly apply to model size, which is in line with the scaling behaviour observed in current time series foundational models (Woo et al., 2024; Goswami et al., 2024). Given that performance generally improves across all models with increasing data size, we hypothesise that scaling the model size could prove beneficial with even larger pre-training corpora.

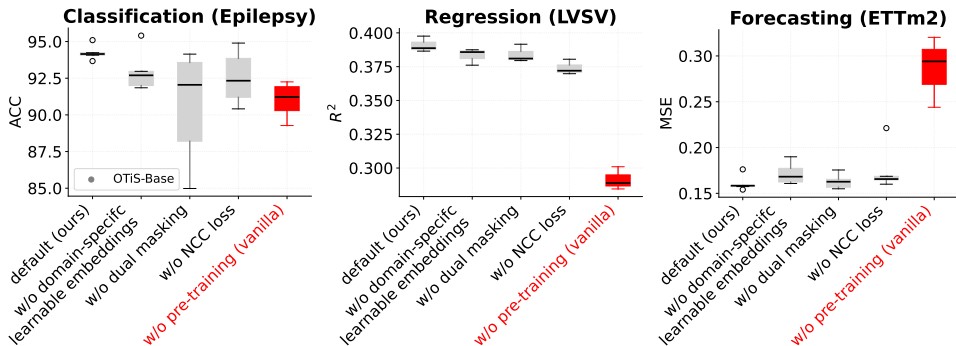

Figure 5: Ablation study on key components of `OTiS`. Downstream performance is analysed across 5 seeds. A leave-one-out approach is used to evaluate the influence of each component. The default setting, that includes all components, demonstrates superior model capabilities across tasks.

## 4.6 ABLATION STUDY

We perform an ablation study to analyse the impact of `OTiS`' key components: the domain-specific tokeniser, dual masking strategy, and normalised cross-correlation (NCC) loss. As shown in Figure 5, the best and most robust performance is achieved when all components are used during pre-training.

Replacing the domain-specific variate embeddings with domain-agnostic embeddings (i.e. learnable embeddings shared across all domains) consistently led to inferior performance across all tasks, demonstrating the importance of capturing domain-specific data characteristics during tokenisation. Switching from dual masking to random masking resulted in performance degradation, although the impact was less notable for generative tasks than for discriminative tasks. We hypothesise that the NCC loss already captures temporal causality, which is particularly crucial for generative tasks like forecasting. Overall, removing the NCC loss caused performance declines across all downstream tasks, emphasising the role of long-range dependencies for general time series understanding.

## 5 DISCUSSION & CONCLUSION

In this study, we explore the problem of effective pre-training on heterogeneous time series corpora. Time series vary substantially across domains, e.g. with respect to inter-variate relationships and temporal dynamics, rendering generalisable feature extraction from multi-domain time series difficult. To address this issue, we present `OTiS`, an **o**pen model for general **ti**me **s**eries analysis, specifically designed to handle multi-domain heterogeneity. Our novel multi-domain pre-training paradigm, including a domain-specific tokeniser with learnable signatures, a dual masking strategy, and a normalised cross-correlation (NCC) loss, enables `OTiS` to extract generalisable time series features.

In extensive experiments, we demonstrate that `OTiS` generalises well across 15 diverse downstream applications spanning 8 distinct domains, achieving competitive performance with both specialised and general state-of-the-art (SOTA) models. In a qualitative analysis, we further show that `OTiS` unifies time series from diverse domains in a meaningful latent space, while preserving low-level semantics of a domain including the inter-variate relationships. Thereby, our work establishes a strong foundation for future advancements in interpretable and general time series analysis.

**Limitations.** While `OTiS` outperforms SOTA models across 10 tasks, our experiments in low-data regimes suggest that larger pre-training corpora could further enhance its performance. Unlike in NLP and CV, where large datasets are curated from web-crawled data, foundational models in time series, including `OTiS`, still rely on manually curated datasets. Future work could explore fully automatic pipelines, e.g. using embedding similarity, to filter and rebalance multi-domain time series from the web. `OTiS` could further benefit from processing domain signatures during inference, potentially unlocking zero-shot capabilities, similarly to those seen in foundational models in NLP and CV.

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

## A LARGE MULTI-DOMAIN PRE-TRAINING CORPUS

In this section, we present an overview of our large and diverse pre-training corpus. The corpus consists of publicly available data spanning eight domains, with a total of $640,187$ samples and 11 billion time points. In the following, we provide a detailed breakdown of the domains and the datasets they encompass. Note that we apply channel-wise standard normalisation to the datasets unless otherwise specified.

**ECG.** The MIMIC-IV-ECG dataset (Gow et al., 2023) contains diagnostic 10-second, 12-lead ECG recordings sampled at a frequency of 500 Hz. While the entire dataset comprises $800,035$ samples, we include only the first half of the recordings available in the database, preventing the ECG data from predominating in the pre-training corpus. To remove the baseline drift from the ECG data, we use the asymmetric least square smoothing technique (Zhang et al., 2010). Note that we apply standard normalisation separately to the Einthoven, Goldberger, and Wilson leads.

**Temperature.** The Deutscher Wetterdienst (DWD) dataset (Wetterdienst, 2024) contains hourly air temperature measurements from 629 weather stations across Germany. Since the recording length varies significantly, ranging from 763 to $1,148,290$ hours per station, we split the data into chunks of 720 hours (approximately one month).

**Audio.** The AudioSet dataset (Gemmeke et al., 2017) contains 10-second YouTube clips for audio classification, featuring 527 types of audio events that are weakly annotated for each clip. The full training set includes a class-wise balanced subset (AudioSet-20K, $22,176$ clips) and an unbalanced (AudioSet-2M $2,042,985$ clips) set. For our pre-training corpus, we use the balanced AudioSet-20K, which contains $3,491$ mono and $16,123$ stereo recordings, all sampled at 44.1 kHz.

**Electromechanics.** The FD-A dataset (Lessmeier et al., 2016) collects vibration signals from rolling bearings in a mechanical system for fault detection purposes. Each sample consists of $5,120$ timestamps, indicating one of three mechanical device states. Note that the FD-B dataset is similar to FD-A but includes rolling bearings tested under different working conditions, such as varying rotational speeds.

**EEG.** The TDBrain dataset (Van Dijk et al., 2022) includes raw resting-state EEG data from $1,274$ psychiatric patients aged 5 to 89, collected between 2001 and 2021. The dataset covers a range of conditions, including Major Depressive Disorder (426 patients), Attention Deficit Hyperactivity Disorder (271 patients), Subjective Memory Complaints (119 patients), and Obsessive-Compulsive Disorder (75 patients). The data was recorded at 500 Hz using 26 channel EEG-recordings, based on the 10-10 electrode international system.

The SEED dataset (Zheng & Lu, 2015) contains EEG data recorded under three emotional states: positive, neutral, and negative. It comprises EEG data from 15 subjects, with each subject participating in experiments twice, several days apart. The data is sampled at 200 Hz and recorded using 62 channel EEG-recordings, based on the 10-20 electrode international system.

For simplicity, we only consider the 19 channels common to both datasets, i.e. the channels that correspond to the 10-20 electrode international system.

**Banking.** The NN5 competition dataset (Taieb et al., 2012) consists of daily cash withdrawals observed at 111 randomly selected automated teller machines across various locations in England.

**Economics.** The FRED-MD dataset (McCracken & Ng, 2016) contains 107 monthly time series showing a set of macro-economic indicators from the Federal Reserve Bank of St Louis. The data was extracted from the FRED-MD database.

The Exchange dataset (Lai et al., 2018) records the daily exchange rates of eight different nations, including Australia, Great Britain, Canada, Switzerland, China, Japan, New Zealand, and Singapore, ranging from 1990 to 2016.

## B    BENCHMARK DETAILS

To assess the utility of `OTiS` in real-world settings, we conduct experiments on three key use cases in time series analysis: classification, regression, and forecasting. For classification, we perform binary epilepsy detection using EEG (Epilepsy 2001), multi-class fault detection in rolling bearings from vibration signals (FD-B 2016), multi-class hand-gesture classification with accelerometer signals (Gesture 2009), and multi-class muscular disease classification using electromyographie (EMG 2000). For regression, we predict five imaging-derived cardiac phenotypes from 12-lead ECG (LVEDV, LVESV, LVSV, LVEF, LVM 2020). For forecasting, we predict electricity transformer temperature (ETT 2021), weather (Weather 2024), and electricity consumption (Electricity 2024). We adhere to the established data splitting and evaluation procedures for the classification (Zhang et al., 2022), regression (Turgut et al., 2023), and forecasting (Zhou et al., 2021) tasks. We provide an overview of the datasets and the baselines used to benchmark our model in Table 4 and Table 5, respectively.

Table 4: Summary of all datasets used for benchmarking, including evaluation metrics, domains, and dataset details.

| Task | Metric | Dataset | | | | | |
|------|--------|---------|------|---------|----------|------------|-----------|
| | | Domain | Name | Samples | Variates $V_S$ | Time points | Frequency |
| Classification | ACC | EEG | Epilepsy 2001 | $11,500$ | 1 | 178 | 174 Hz |
| | | | TUEV 2016 | $112,237$ | 19 | $1,000$ | 200 Hz |
| | | Electromechanics | FD-B 2016 | $13,640$ | 1 | $5,120$ | 64 kHz |
| | | Acceleration | Gesture 2009 | 560 | 3 | 206 | 100 Hz |
| | | EMG | EMG 2000 | 204 | 1 | $1,500$ | 4 kHz |
| Regression | $R^2$ | ECG | UK BioBank 2015 | $18,926$ | 12 | $5,000$ | 500 Hz |
| Forecasting | MSE | Energy | ETTh1 2021 | 1 | 7 | $17,420$ | (hourly) 278 $\mu$Hz |
| | | | ETTh2 2021 | 1 | 7 | $17,420$ | (hourly) 278 $\mu$Hz |
| | | | ETTm1 2021 | 1 | 7 | $69,680$ | (minutely) 1.1 mHz |
| | | | ETTm2 2021 | 1 | 7 | $69,680$ | (minutely) 1.1 mHz |
| | | Weather | Weather 2024 | 1 | 21 | $52,696$ | (minutely) 2.8 mHz |
| | | Electricity | Electricity 2024 | 321 | 1 | $26,304$ | (hourly) 278 $\mu$Hz |

## C    EXPERIMENT DETAILS

### C.1    MODEL VARIANTS

To explore the scaling laws with respect to the model size, we provide `OTiS` in three variants, as summerised in Table 6.

### C.2    PRE-TRAINING & FINE-TUNING PARAMETERS

We provide the hyperparameters used to pre-train all variants of `OTiS` in Table 7. The hyperparameters used to fine-tune our models for the classification, regression, and forecasting tasks are provided in Table 8, 9, and 10, respectively.

## D    COMPUTATION COSTS

We provide an overview of the computational resources used to train `OTiS` in Table 11.

Table 5: Summary of all baseline models used for benchmarking, including pre-training details, domain adaptation methods, and architectural choices. CL, MDM, and GPT denote contrastive learning, masked data modelling, and generative pre-training, respectively.

| Task | Model | Pre-training | | Domain adaptation | Architecture |
|---|---|---|---|---|---|
| | | Method | Dataset | | |
| Classification | SimCLR 2020 | CL | SleepEEG* 2000 | Fine-tuning | 1D-CNN |
| | TimesNet 2022a | – | Target | Fine-tuning | 2D-CNN |
| | CoST 2022 | CL | SleepEEG* 2000 | Fine-tuning | 1D-CNN |
| | TS2Vec 2022 | CL | SleepEEG* 2000 | Fine-tuning | 1D-CNN |
| | TF-C 2022 | CL | SleepEEG* 2000 | Fine-tuning | Transformer |
| | Ti-MAE 2023 | MDM | SleepEEG* 2000 | Fine-tuning | Transformer |
| | SimMTM 2024 | MDM | SleepEEG* 2000 | Fine-tuning | Transformer |
| Regression | iTransformer 2023 | – | Target | Fine-tuning | Transformer |
| | ViT 2023 | – | Target | Fine-tuning | Transformer |
| | MAE 2023 | MDM | Target | Fine-tuning | Transformer |
| | CM-AE 2023 | MDM and CL | Target | Fine-tuning | 1D-CNN |
| | MMCL 2023 | MDM and CL | Target | Fine-tuning | Transformer |
| Forecasting | N-BEATS 2019 | – | Target | Fine-tuning | Non-Linear Model |
| | Autoformer 2021 | – | Target | Fine-tuning | Transformer |
| | TimesNet 2022a | – | Target | Fine-tuning | 2D-CNN |
| | DLinear 2023 | – | Target | Fine-tuning | Linear Model |
| | PatchTST 2023 | MDM | Target | Fine-tuning | Transformer |
| | Time-LLM 2023 | GPT | † | Fine-tuning | Transformer |
| | GPT4TS 2023 | GPT | ‡ | Fine-tuning | Transformer |
| | MOMENT 2024 | MDM | TSP° 2024 | Fine-tuning | Transformer |
| | MOIRAI 2024 | MDM | LOTSA◁ 2024 | Zero-shot | Transformer |

* $371,055$ uni-variate, 2-seconds EEG recordings sampled at a frequency of 100 Hz.

† Llama-7B 2023, pre-trained on $1.4$ trillion text tokens, is used as backbone.

‡ GPT2 2018, pre-trained on 10 billion text tokens, is used as backbone.

° Time Series Pile (TSP) contains 13 million samples and $1.23$ billion time points from 13 domains.

◁ Large-Scale Open Time Series Archive (LOTSA) contains more than 4 million samples and 27 billion time points from 9 domains.

Table 6: Details of model variants.

| Model | Layers | Hidden size $D$ | MLP size | Heads | $d_{kv}$ | Parameters |
|---|---|---|---|---|---|---|
| OTiS-Base | 12 | 192 | 768 | 3 | 64 | 8 M |
| OTiS-Large | 18 | 384 | 1536 | 6 | 64 | 44 M |
| OTiS-Huge | 24 | 576 | 2304 | 8 | 72 | 131 M |

Table 7: Hyperparameters used for pre-training. Pre-training is performed on 4 NVIDIA A100-80GB GPUs. A cosine learning rate scheduler is applied with a $10\%$ warmup. All OTiS configurations use a shallow decoder with 2 M parameters, consisting of 4 layers with a hidden size of 160, an MLP with size 640, and 5 heads.

| Model | Epochs | Batch size | Base LR | LR decay | NCC $\lambda$ | Mask ratio $\rho$ | Weight decay |
|---|---|---|---|---|---|---|---|
| OTiS-Base | 200 | 5120 | 3e-5 | cosine | 0.1 | 0.75 | 0.10 |
| OTiS-Large | 200 | 3328 | 1e-5 | cosine | 0.1 | 0.75 | 0.15 |
| OTiS-Huge | 200 | 2880 | 3e-6 | cosine | 0.1 | 0.75 | 0.05 |

# E  DOMAIN SIGNATURE ANALYSIS

To analyse the domain signatures, we reduce the dimensionality of the domain-specific variate embeddings by employing a principal component analysis (PCA). Our analysis shows that OTiS unifies time series from diverse domains into a meaningful latent space, while accurately capturing the inter-variate relationships within a domain.

Table 8: Hyperparameters used for fine-tuning the classification tasks on a single NVIDIA RTX A6000-48GB GPU. A cosine learning rate scheduler is applied with a $10\%$ warmup.

| Dataset | Model | Epochs | Batch size | Base LR | Drop path | Layer decay | Weight decay | Label smoothing |
|---|---|---|---|---|---|---|---|---|
| Epilepsy | OTiS-Base | 75 | 32 | 1e-3 | 0.2 | 0.75 | 0.2 | 0.1 |
| | OTiS-Large | 75 | 32 | 3e-3 | 0.2 | 0.50 | 0.1 | 0.1 |
| | OTiS-Huge | 75 | 32 | 3e-3 | 0.0 | 0.75 | 0.2 | 0.2 |
| FD-B | OTiS-Base | 75 | 32 | 3e-4 | 0.0 | 0.75 | 0.1 | 0.1 |
| | OTiS-Large | 75 | 32 | 1e-3 | 0.1 | 0.75 | 0.1 | 0.2 |
| | OTiS-Huge | 75 | 32 | 3e-4 | 0.1 | 0.75 | 0.2 | 0.1 |
| Gesture | OTiS-Base | 75 | 32 | 3e-3 | 0.2 | 0.50 | 0.1 | 0.1 |
| | OTiS-Large | 75 | 32 | 3e-3 | 0.2 | 0.75 | 0.1 | 0.0 |
| | OTiS-Huge | 75 | 32 | 1e-2 | 0.0 | 0.75 | 0.1 | 0.1 |
| EMG | OTiS-Base | 75 | 32 | 1e-3 | 0.2 | 0.75 | 0.1 | 0.2 |
| | OTiS-Large | 75 | 32 | 3e-3 | 0.1 | 0.75 | 0.2 | 0.1 |
| | OTiS-Huge | 75 | 32 | 3e-3 | 0.1 | 0.75 | 0.2 | 0.2 |

Table 9: Hyperparameters used for fine-tuning the regression tasks on a single NVIDIA RTX A6000-48GB GPU. A cosine learning rate scheduler is applied with a $10\%$ warmup.

| Dataset | Model | Epochs | Batch size | Base LR | Drop path | Layer decay | Weight decay |
|---|---|---|---|---|---|---|---|
| UK BioBank | OTiS-Base | 50 | 192 | 3e-4 | 0.2 | 0.75 | 0.1 |
| | OTiS-Large | 50 | 160 | 1e-4 | 0.2 | 0.75 | 0.1 |
| | OTiS-Huge | 50 | 200 | 1e-4 | 0.2 | 0.75 | 0.1 |

Table 10: Hyperparameters used for fine-tuning the forecasting tasks. A cosine learning rate scheduler is applied with a $10\%$ warmup.

| Dataset | Model | Epochs | Batch size | Base LR | NCC $\lambda$ | Weight decay |
|---|---|---|---|---|---|---|
| ETTh1 | OTiS-Base | 1000 | 1 | 1e-0 | 0.1 | 0.15 |
| | OTiS-Large | 1000 | 1 | 1e-1 | 0.2 | 0.15 |
| | OTiS-Huge | 1000 | 1 | 3e-1 | 0.1 | 0.15 |
| ETTh2 | OTiS-Base | 1000 | 1 | 1e-0 | 0.2 | 0.25 |
| | OTiS-Large | 1000 | 1 | 1e-1 | 0.1 | 0.25 |
| | OTiS-Huge | 1000 | 1 | 3e-1 | 0.0 | 0.25 |
| ETTm1 | OTiS-Base | 1000 | 1 | 3e-1 | 0.2 | 0.25 |
| | OTiS-Large | 1000 | 1 | 3e-1 | 0.2 | 0.25 |
| | OTiS-Huge | 1000 | 1 | 1e-1 | 0.1 | 0.15 |
| ETTm2 | OTiS-Base | 1000 | 1 | 3e-1 | 0.1 | 0.25 |
| | OTiS-Large | 1000 | 1 | 1e-1 | 0.2 | 0.25 |
| | OTiS-Huge | 1000 | 1 | 3e-1 | 0.2 | 0.25 |
| Weather | OTiS-Base | 1000 | 1 | 3e-1 | 0.2 | 0.25 |
| | OTiS-Large | 1000 | 1 | 3e-1 | 0.2 | 0.15 |
| | OTiS-Huge | 1000 | 1 | 1e-1 | 0.2 | 0.05 |
| Electricity | OTiS-Base | 250 | 32 | 3e-2 | 0.0 | 0.25 |
| | OTiS-Large | 250 | 32 | 3e-2 | 0.0 | 0.15 |
| | OTiS-Huge | 250 | 32 | 3e-2 | 0.2 | 0.15 |

Table 11: Computational resources used to pre-train OTiS. Note that fine-tuning and inference of all OTiS variants on downstream applications were performed using a single NVIDIA RTX A6000-48GB and 32 CPUs.

| Model | Parameters | Power consumption | CPU count | GPU Count | GPU Hours | GPU Type |
|---|---|---|---|---|---|---|
| OTiS-Base | 8 M | 700 W* | 128 | 4 | 115† | NVIDIA A100-80GB |
| OTiS-Large | 44 M | 800 W* | 128 | 4 | 154† | NVIDIA A100-80GB |
| OTiS-Huge | 131 M | 960 W* | 128 | 4 | 219† | NVIDIA A100-80GB |

\* Total power consumption across all GPUs.

† Total hours across all GPUs.

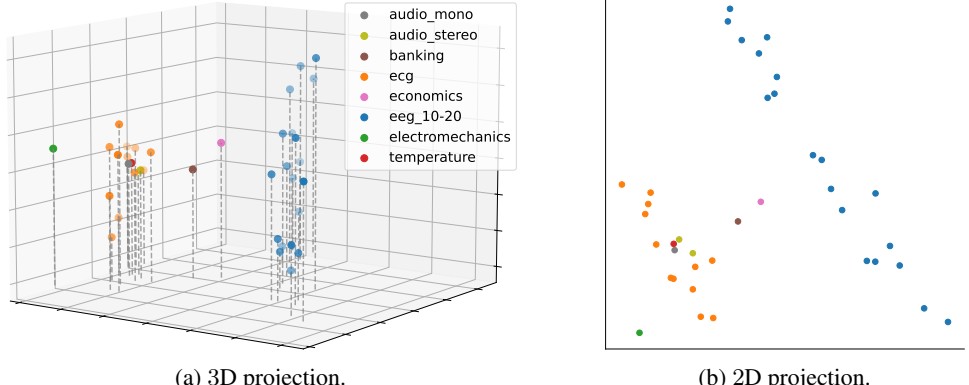

(a) 3D projection.                (b) 2D projection.

Figure 6: PCA projections of the domain-specific variate embeddings learned during pre-training. OTiS unifies time series from diverse domains in a meaningful latent space, while correctly encoding the inter-variate relationships within a domain. Mono (●) and stereo (●) audio-specific embeddings cluster closely together, as do those for banking (●) and economics (●). Clear separation is observed for EEG-specific embeddings (●), while also ECG-specific embeddings (●) form a tight cluster.

## E.1 INTER-DOMAIN ANALYSIS

A visualisation of all domain-specific variate embeddings learned during pre-training is provided in Figure 6. We find that OTiS learns a meaningful latent space, where embeddings of domains with shared high-level semantics cluster closely together.

## E.2 INTRA-DOMAIN ANALYSIS

### E.2.1 ALIGNMENT OF EEG-SPECIFIC VARIATE EMBEDDINGS WITH THE TRUE ELECTRODE LAYOUT

We assume 3D electrode coordinates of the international 10-20 system for EEG recordings (Homan et al., 1987) to be defined in Euclidean space $\mathbb{E}_Y^3$ (see Figure 7a). To determine how well the learned EEG-specific variate embeddings reflect the true electrode layout, we project them into Euclidean space $\mathbb{E}_X^3$ (see Figure 7b), linearly align them with the true 3D electrode coordinates in $\mathbb{E}_Y^3$, and eventually quantify their correlation, as described in the following.

First, we determine the first three principal components of the EEG-specific variate embeddings, thus projecting them into a Euclidean space $\mathbb{E}_X^3$. Then, we perform a multivariate linear regression

$$\mathbf{Y} = \mathbf{1}\beta_0 + \mathbf{X}\mathbf{B} + \epsilon \in \mathbb{R}^{N \times 3} \quad \text{with} \quad \beta_0 \in \mathbb{R}^{1 \times 3}, \mathbf{X} \in \mathbb{R}^{N \times 3}, \mathbf{B} \in \mathbb{R}^{3 \times 3}, \epsilon \in \mathbb{R}^{N \times 3}, \quad (7)$$

where $\mathbf{1} \in \mathbb{R}^{N \times 1}$ is a vector of ones and $N$ denotes the number of electrodes, to align the first three principal components in $\mathbb{E}_X^3$ (here, $\mathbf{X}$) with the 3D electrode coordinates in $\mathbb{E}_Y^3$ (here, $\mathbf{Y}$). Finally, to quantify this very alignment, we determine the coefficient of determination $R^2 \in [0, 1]$. Note that

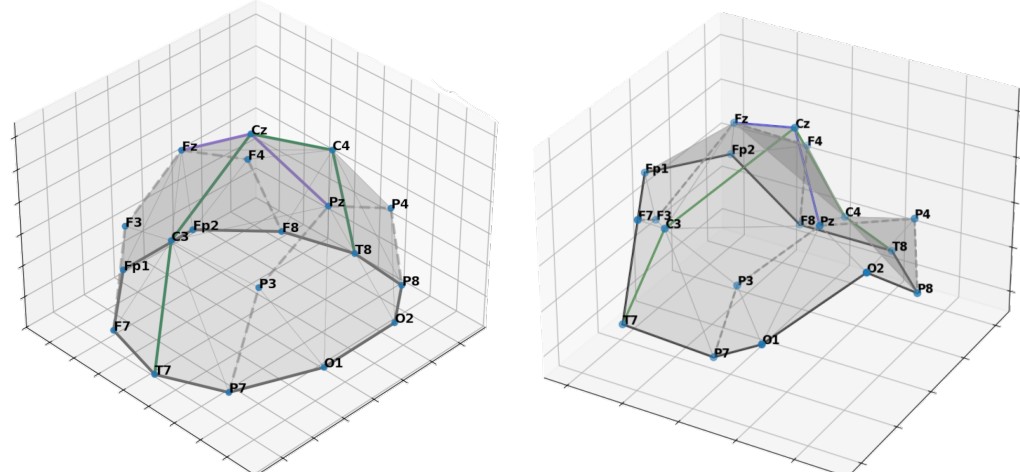

(a) 3D electrode layout of the international 10-20 system for EEG recordings.

(b) First three principal components of the EEG-specific variate embeddings.

Figure 7: Analysis of the variate embeddings for 10-20 system EEG recordings with 19 electrodes learned during pre-training. The label of each coordinate corresponds to the electrode name.

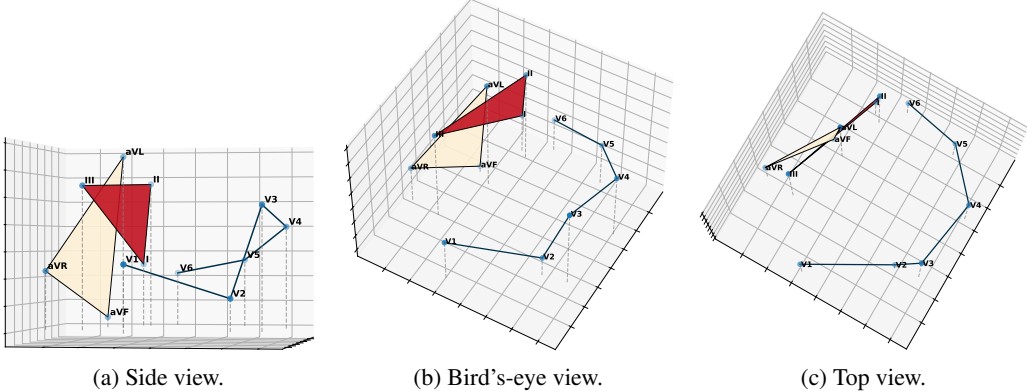

(a) Side view.                    (b) Bird's-eye view.                    (c) Top view.

Figure 8: Principal component analysis of the variate embeddings for standard 12-lead ECG learned during pre-training. Their first three components, shown in (a), (b), and (c), accurately reflect the true physiological structure of ECG leads. The V1-V6 leads, arranged on the rib cage from the sternum to the mid-axillary line, represent a 3D view of the human heart. The I-II-III leads and aVR-aVL-aVF leads, derived from electrodes placed on one foot and both arms, form a planar 2D triangle.

$R^2 = 1$ represents a perfect alignment, i.e. $\mathbb{E}_Y^3 = \mathbb{E}_X^3$, where the first three principal components of the EEG-specific variate embeddings only need to be shifted and scaled to retrieve the true EEG electrode layout (i.e. $\epsilon$ is a zero matrix).

### E.2.2 ADDITIONAL ANALYSES OF DOMAIN-SPECIFIC VARIATE EMBEDDINGS

To further explore OTiS' ability to capture complex inter-variate relationships across domains, we analyse (i) the ECG-specific variate embeddings learned during pre-training and (ii) the Weather-specific variate embeddings learned during fine-tuning. Figure 8 presents a principal components analysis of the ECG-specific variate embeddings. Since these were learned during pre-training, similar to the EEG-specific embeddings discussed in Section 4.4 and Section E.2.1, we focus the following analysis on the Weather-specific variate embeddings learned during fine-tuning.

The central question is whether OTiS can learn domain-specific knowledge - in this case, for the weather domain - from limited data seen only during fine-tuning. To investigate this, we compute

the cosine similarity for all pairs of Weather-specific variate embeddings, as summarised in Figure 9. Note that these embeddings were randomly initialised and learned specifically for the Weather 2024 dataset during the forecasting task described in Section B. The Weather variates span diverse climatological categories, including temperature (T, Tpot, Tdew, Tlog), humidity (rh, VPmax, VPact, VPdef, sh, H2OC), wind (wv, max. wv, wd), radiation (SWDR, PAR, max. PAR), pressure (p, rho), and precipitation (rain, raining). Our analysis demonstrates that `OTiS` effectively captures complex relationships among these distinct climatological indicators, as detailed in the following discussion.

**High positive similarities**   typically indicate relationships within a single climatological category. For example, we observe strong similarities among temperature variates, humidity variates, radiation variates, pressure variates, and precipitation variates. These results are expected, as these variates all describe different aspects of the same category and often fluctuate together. Additionally, subtle variations in the similarity scores reveal how, for instance, dew point temperature (Tdew) depends not only on temperature but also on other factors, such as humidity (rh).

**High negative similarities**   typically represent relationships across climatological categories. For example, consider the inverse relationship between vapor pressure deficit (VPdef) and relative humidity (rh), defined as:

$$\text{VPdef} = \text{SVP}\left(1 - \frac{\text{rh}}{100}\right), \tag{8}$$

where SVP [mBar] denotes the saturation vapor pressure. Our analysis showcases that `OTiS` correctly captures this negative correlation, as well as other relationships across categories. These include the inverse correlation between strong winds (max. wv) and low air pressure (p), and between extended precipitation (raining) and lower incoming radiation (SWDR).

## F   ZERO-SHOT CAPABILITIES

We analyse `OTiS`' zero-shot capabilities across four diverse tasks: binary epilepsy detection using uni-variate EEG (Epilepsy 2001), multi-class fault detection in rolling bearings from uni-variate electromechanics signals (FD-B 2016), multi-class hand-gesture classification with multi-variate accelerometer signals (Gesture 2009), and multi-class muscular disease classification using uni-variate electromyographie (EMG 2000). These datasets vary significantly in domain, number of variates, time points, sampling frequency, and number of classes, highlighting the versatility of our analysis. Details on the datasets can be found in Table 4.

In the zero-shot setting, `OTiS` is evaluated without domain-specific fine-tuning by freezing it after pre-training and using randomly initialised variate embeddings. Since no classification head is employed, the encoder's output tokens are averaged to obtain a global representation for each sample. To create class representations, the global representations of the training samples are averaged separately for each class. For classification, the cosine similarity is computed between each test sample's global representation and the class representations. The class with the highest similarity score is assigned to the test sample. As illustrated in Figure 10, `OTiS` is able to extract distinct representations for different classes, even without domain-specific fine-tuning. This ability translates to zero-shot classification accuracies of $93.70\,\%$ for Epilepsy, $57.87\,\%$ for FD-B, $51.67\,\%$ for Gesture, and $95.12\,\%$ for EMG. A closer examination of the zero-shot latent space for FD-B (Figure 10b) reveals a partial overlap of inputs from classes 1 and 2, which explains the lower zero-shot performance compared to the fine-tuning results (Figure 11b). Similarly, inputs from the eight classes in the Gesture dataset show poor clustering in the zero-shot latent space (Figure 10c), with only slight improvements observed after fine-tuning (Figure 11c). Overall, these quantitative and qualitative zero-shot findings highlight `OTiS`' ability to extract time series features that generalise across domains and tasks, providing a strong foundation for future advancements in general time series analysis.

Additionally, since our pre-training corpus includes time series from EEG (TDBrain 2022 and SEED 2015) and Electromechanics (FD-A 2016), we also evaluate zero-shot performance using the EEG- and Electromechanics-specific variate embeddings learned during pre-training instead of randomly initialised ones. As anticipated, leveraging these learned variate embeddings enhances the quality of the generated representations, resulting in improved zero-shot classification accuracies of $95.18\,\%$ for Epilepsy ($+1.48\,\%$) and $61.32\,\%$ for FD-B ($+3.45\,\%$).

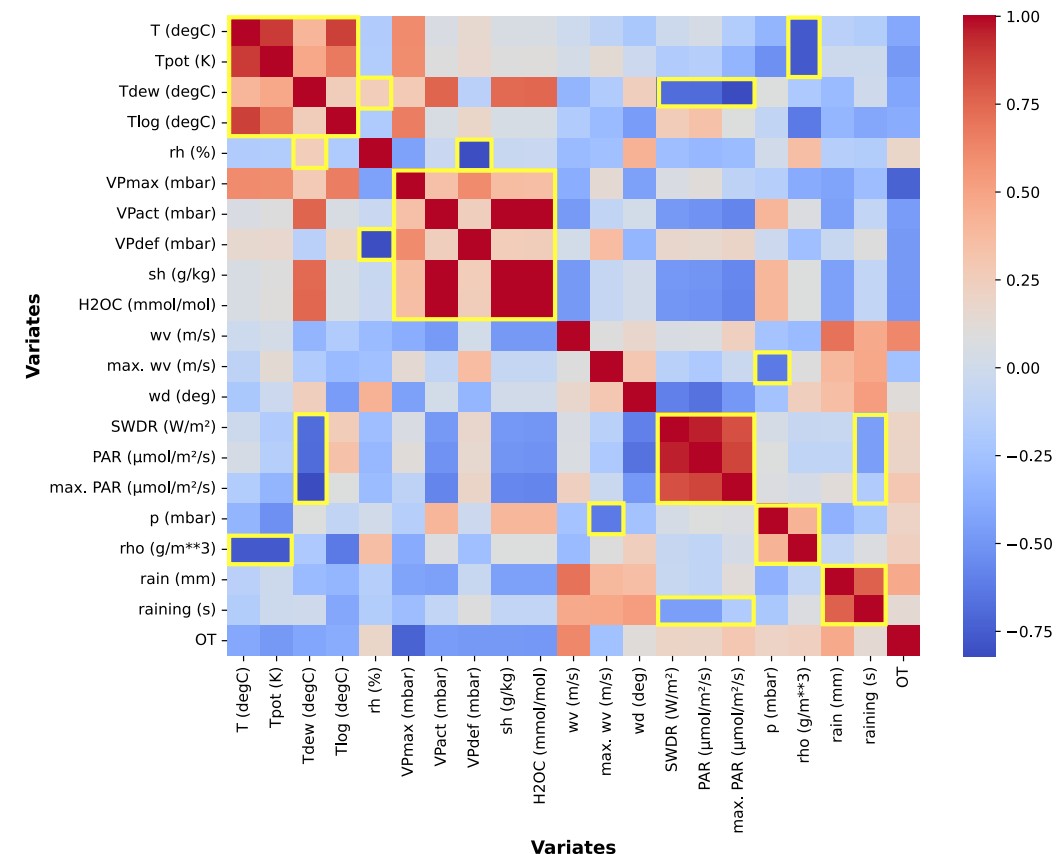

Figure 9: Cosine similarity matrix of Weather-specific variate embeddings. Note that the ordering of the variates was modified for visualisation purposes. Areas with high positive and high negative similarity are exemplary framed in yellow. `OTiS` is capable of capturing non-trivial relationships between climatological indicators of the Weather 2024 dataset.

## G ADDITIONAL ABLATION STUDIES

### G.1 DUAL MASKING STRATEGY

In order to enhance `OTiS`' foundational capabilities for general time series analysis, we incorporate a dual masking strategy in our pre-training strategy, as described in Section 3.4. Specifically, we select between two masking schemes during pre-training: random masking (randomly masking across variate and temporal dimension) and post-fix masking (masking the second half of the temporal dimension). To determine the optimal balance between these two schemes, we examine the impact of different compositions of the dual masking strategy across distinct use cases in time series analysis. Our analysis reveals that a combination of 75 % random masking and 25 % post-fix masking consistently yields the best downstream performance across all tasks, as illustrated in Figure 12.

### G.2 PRE-TRAINING STRATEGY

To explore whether domain-specific pre-training is beneficial over pre-training on diverse time series across domains, we analyse different training strategies for EEG event type classification on the TUEV 2016 dataset, as summarised in Table 12. In particular, we compare `OTiS` against specialised and general baseline models that are particularly designed for EEG analysis. The baselines include i) specialised models that are randomly initialised and trained fully supervised on the target data, ii) self-supervised models pre-trained and fine-tuned on the target data, and iii) foundation models pre-trained on large time series corpora and fine-tuned on the target data. The specialised baselines include ST-Transformer (Song et al., 2021) (Transformer), CNN-Transformer (Peh et al., 2022) (CNN and

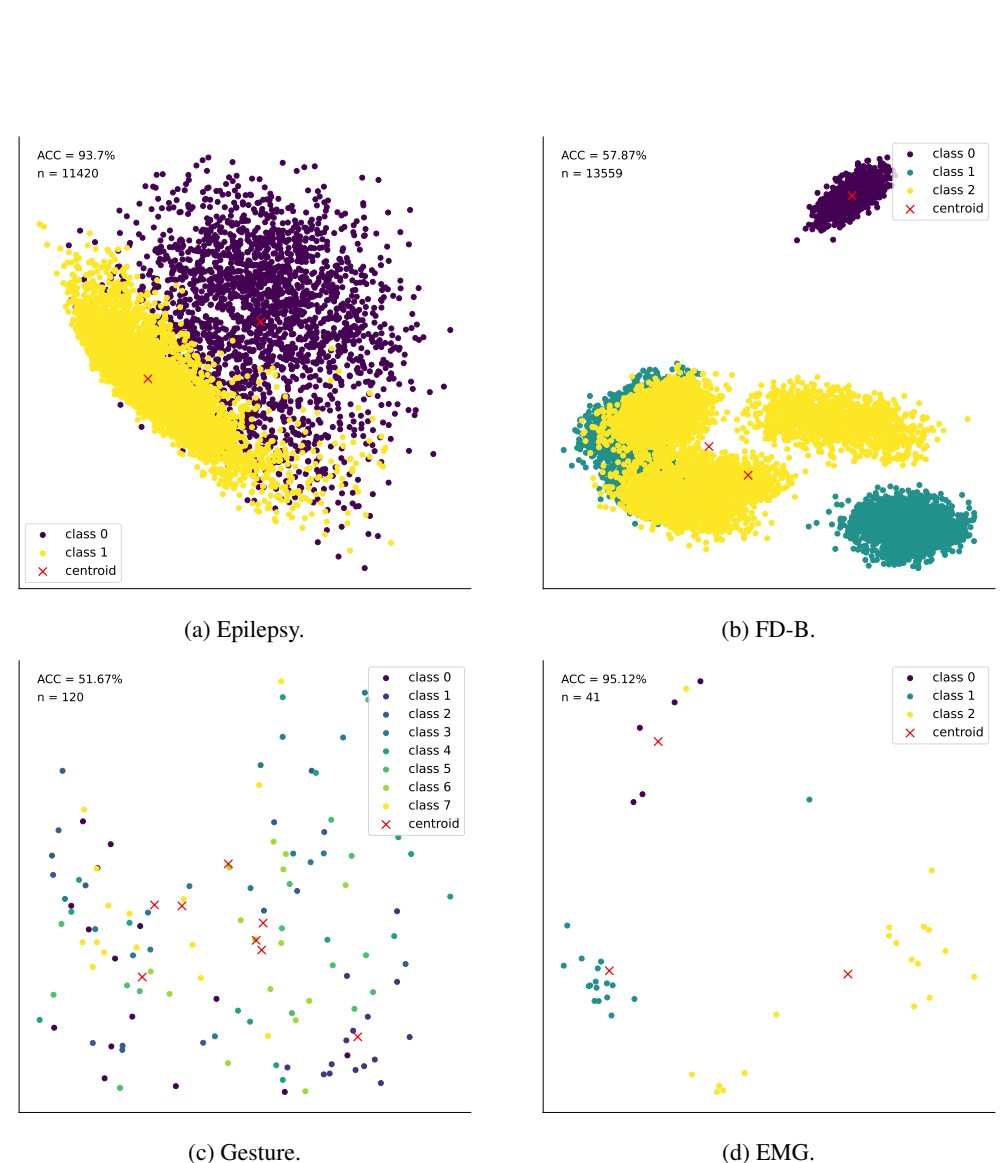

(a) Epilepsy.

(b) FD-B.

(c) Gesture.

(d) EMG.

Figure 10: First two principal components of the *zero-shot* representations generated by `OTiS`-Base across four datasets. In this setup, `OTiS` is frozen after pre-training and randomly initialised variate embeddings are utilised. As no classification head is employed, the output tokens of the encoder are averaged to obtain a global representation. `OTiS` extracts distinct representations for different inputs, even across domains and tasks, highlighting its potential for general time series analysis.

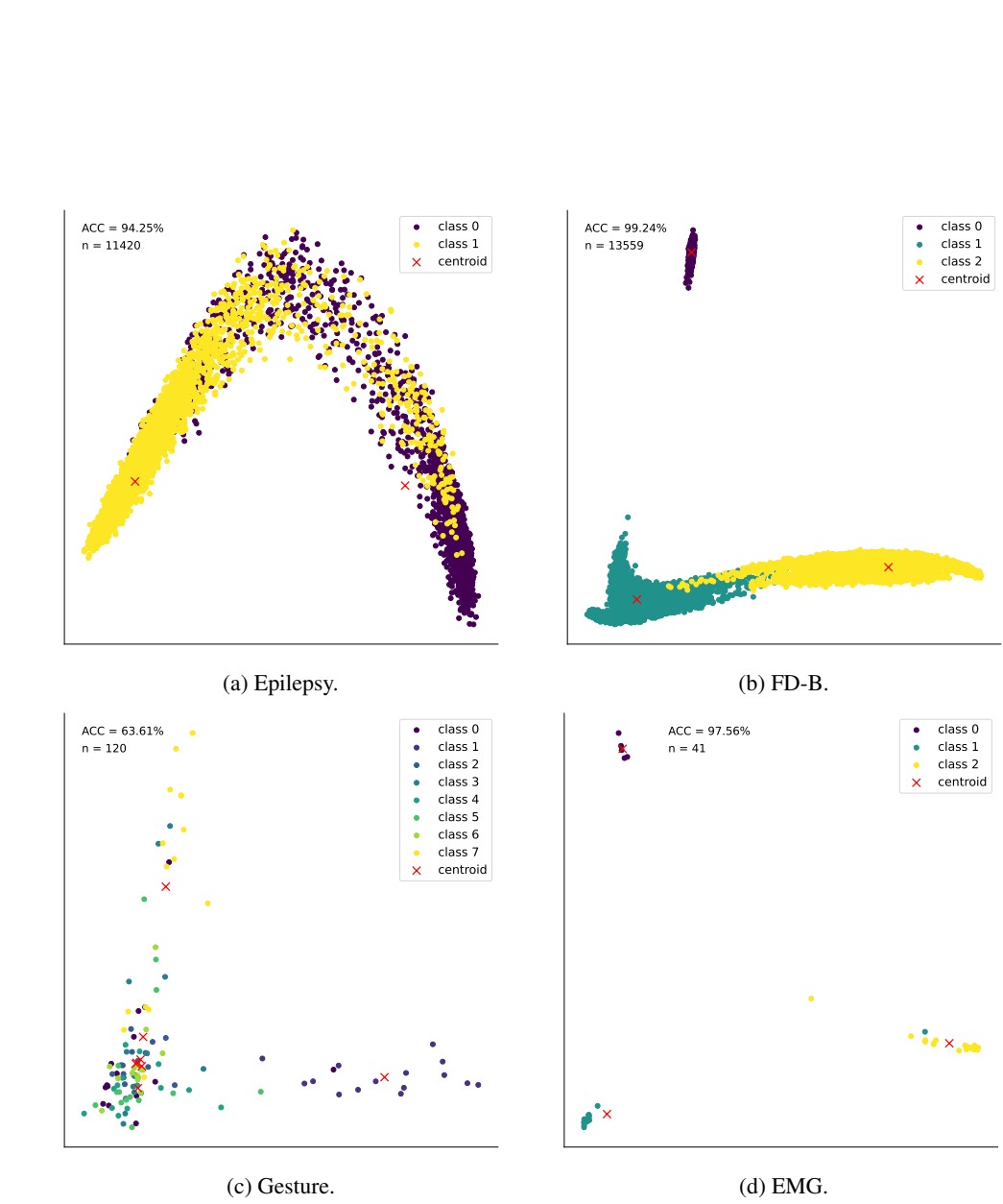

(a) Epilepsy.

(b) FD-B.

(c) Gesture.

(d) EMG.

Figure 11: First two principal components of the *fine-tuned* representations generated by OTiS-Base across four datasets. Fine-tuning enabels OTiS to form tight clusters for distinct classes, highlighting its effective adaptation to specific tasks regardless of their domain.

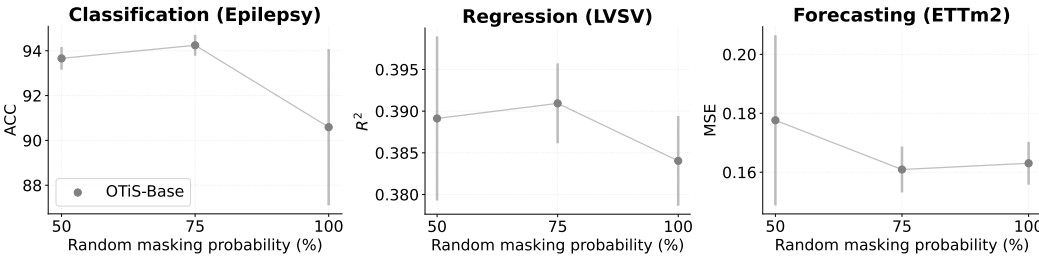

Figure 12: **Ablation study on the composition of the dual masking strategy.** Error bars indicate the standard deviation across 5 seeds. A combination of 75 % random masking and 25 % post-fix masking during pre-training consistently yields the best downstream performance across tasks.

Transformer), FFCL (Li et al., 2022) (CNN and LSTM), and SPaRCNet (Jing et al., 2023) (1D-CNN). Self-supervised baselines include ContraWR (Yang et al., 2023) (2D-CNN). The foundation models include BIOT (Yang et al., 2024) (Transformer, pre-trained on 6 EEG datasets with over 5 million samples and 13,000 recording hours) and LaBraM (Jiang et al., 2024) (Transformer, pre-trained on 16 EEG datasets with a total of 2,500 recording hours). Similar to `OTiS`, both foundation models are pre-trained leveraging masked data modelling. However, the EEG datasets in our pre-training corpus consist of only 125 recording hours (90 hours in TDBrain 2022 and 35 hours in SEED 2015), which is substantially lower than the EEG corpora used by baselines.

The experiments show that pre-trained models (ii) and (iii) outperform the fully-supervised models (i). Notably, for `OTiS`, pre-training exclusively on EEG data does not yield to improved downstream performance compared to general pre-training across diverse time series. Moreover, competitive downstream results can be achieved even without incorporating explicit domain knowledge, as shown by `OTiS` with randomly initialised variate-embeddings before fine-tuning, i.e. `OTiS-Base`$_\text{w/ rVE}$. Our randomly initialised model, `OTiS-Base`$_\text{w/p pre-training}$, outperforms all other specialised models and performs on par with the self-supervised SPaRCNet and foundational BIOT, suggesting an efficient interplay between the domain-specific tokeniser and Transformer backbone. Overall, these results from EEG event type classification highlight that pre-training across domains generally enhances the quality of representations generated by `OTiS`, translating to superior downstream performance.

Table 12: **Ablation study on pre-training strategies for EEG event type classification on the TUEV 2016 dataset.** Mean and standard deviation is reported across 5 seeds set during fine-tuning. Best score in **bold**, second best underlined. All baselines are specifically tailored for EEG analysis, including foundation models ([‡]) pre-trained on large EEG corpora and fine-tuned on the target data.

| Methods | Parameters | Balanced ACC ↑ | Cohen's Kappa ↑ | Weighted F1 ↑ |
|---|---|---|---|---|
| ST-Transformer 2021 | 3.5 M | $0.3984 \pm 0.0228$ | $0.3765 \pm 0.0306$ | $0.6823 \pm 0.0190$ |
| CNN-Transformer 2022 | 3.2 M | $0.4087 \pm 0.0161$ | $0.3815 \pm 0.0134$ | $0.6854 \pm 0.0293$ |
| FFCL 2022 | 2.4 M | $0.3979 \pm 0.0104$ | $0.3732 \pm 0.0188$ | $0.6783 \pm 0.0120$ |
| SPaRCNet 2023 | 0.79 M | $0.4161 \pm 0.0262$ | $0.4233 \pm 0.0181$ | $0.7024 \pm 0.0104$ |
| ContraWR 2023 | 1.6 M | $0.4384 \pm 0.0349$ | $0.3912 \pm 0.0237$ | $0.6893 \pm 0.0136$ |
| BIOT [‡] 2024 | 3.2 M | $0.5281 \pm 0.0225$ | $0.5273 \pm 0.0249$ | $0.7492 \pm 0.0082$ |
| LaBraM [‡] 2024 | 369 M | $\mathbf{0.6616 \pm 0.0170}$ | $\mathbf{0.6745 \pm 0.0195}$ | $\mathbf{0.8329 \pm 0.0086}$ |
| `OTiS-Base`$_\text{w/o pre-training}$[*] | 8 M | $0.5361 \pm 0.0350$ | $0.5183 \pm 0.0316$ | $0.7642 \pm 0.0157$ |
| `OTiS-Base`$_\text{EEG}$[†] | 8 M | $0.5562 \pm 0.0106$ | $0.5504 \pm 0.0204$ | $0.7784 \pm 0.0095$ |
| `OTiS-Base`$_\text{EEG w/ rVE}$[†▷] | 8 M | $0.5413 \pm 0.0302$ | $0.5631 \pm 0.0299$ | $0.7860 \pm 0.0120$ |
| `OTiS-Base` | 8 M | $0.5743 \pm 0.0257$ | $0.5913 \pm 0.0146$ | $0.8004 \pm 0.0071$ |
| `OTiS-Base`$_\text{w/ rVE}$[▷] | 8 M | $0.5728 \pm 0.0134$ | $0.5772 \pm 0.0281$ | $0.7922 \pm 0.0127$ |

[*] Model was randomly initialised and trained fully supervised.

[†] Model was pre-trained only with the EEG data of our pre-training corpus.

[▷] Variate embeddings (VE) are randomly initialised before for fine-tuning.

# H   FORECAST VISUALISATION

We visualise the performance of our model on 6 forecasting benchmarks in Figure 13.

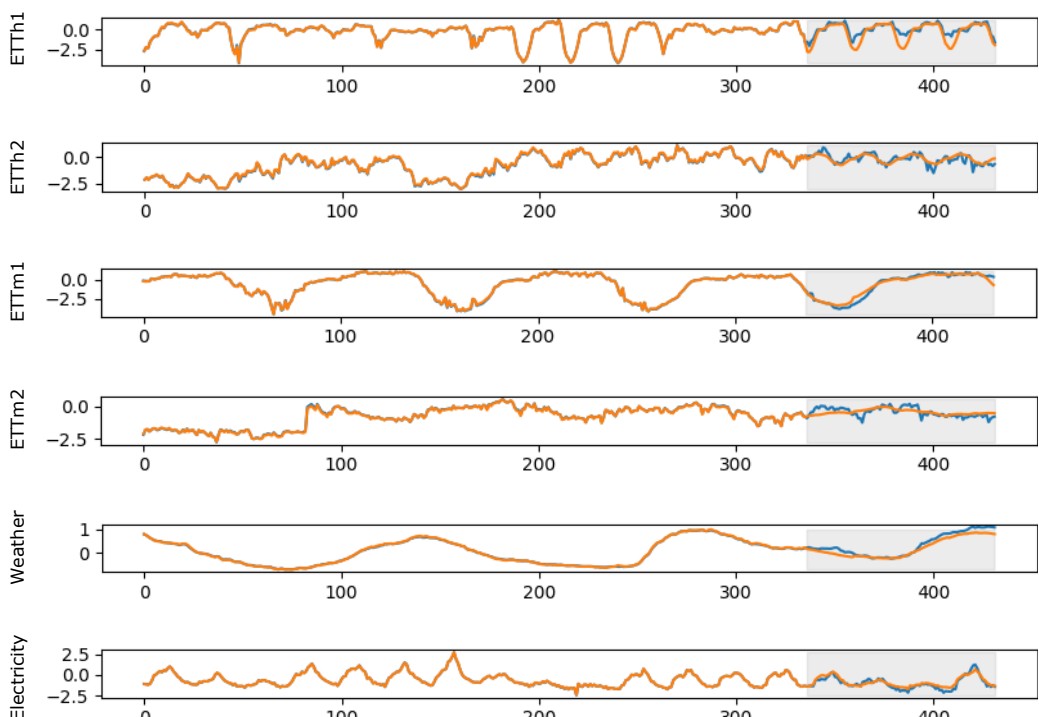

Figure 13: Visualisation of `OTiS`-Base forecast predictions on 6 benchmark datasets. A forecasting horizon of 96 time points is predicted from the past 336 time points. Ground truth in blue, prediction in orange. Areas highlighted in grey are not visible to the model.

