# OpenReview forum: "Towards Generalisable Time Series Understanding Across Domains"
_ICLR.cc/2025/Conference — Submitted to ICLR 2025_

### Official Review · Reviewer_tXLU · 2024-10-27

**Soundness:** 3
**Presentation:** 2
**Contribution:** 3
**Rating:** 5
**Confidence:** 3

**Summary:**

The paper presents OTiS, a deep model pre-trained on a large corpus (11B) for general time series analysis. In this paper, the authors highlight the challenge of heterogeneity when applying self-supervised pre-training on time series. An MAE-style pre-training method is adopted to obtain a general tokenizer for multivariate time series, and then different task heads are introduced to complete time series analysis tasks.  The model demonstrates strong performance across 15 diverse applications, including time series classification, regression, and forecasting.

**Strengths:**

1. This paper researches an important question about generalizable time series understanding across diverse domains.
2. This work presents a large pre-training corpus, which can be greatly beneficial if the datasets are released.
3. The method exhibits promising results in handling multivariate time series analysis by leveraging variate and domain signatures.

**Weaknesses:**

1. My major concern is about the novelty of the proposed method: The design of the encoder/decoder is very identical to MAE. Is there any adaptation for the time series modality? For example, considering the inherent reconstruction difficulties of time series and adjusting the mask ratio compared with the vision modality?
2. About the model design towards generalizable time series understanding: As the authors mention an important challenge of heterogeneity, I am slightly unconvinced that a shared unified patch embedding/projector can reflect different semantics among variates and domains, even if the patch is completely the same. Prior to this, Moirai adopted different patch sizes for different frequencies, will it further enhance OTiS?
3. This work adopts learnable embeddings as variate/domain signatures. I am convinced that the signatures can "distinguish" them, but how can they explicitly "capture inter-variate relationships"? This approach may also limit the generalization scope as the learned signatures do not apply to unseen variates/domains during inference.
4. About the experiments: Results of classification are not compared with supervised, trained deep models, for example, TimesNet and ModernTCN. For the regression rask, can you introduce some variate-centric models into this baseline, such as iTransformer? As for forecasting, the average improvement does not seem significant compared with PatchTST. Also, can you provide some explanations about Table 3 why OTiS has a significant improvement on some datasets (such as ETTh2) and a great degeneration on similar datasets like ETTh1?
5. A minor suggestion: the name "dual masking strategy" can be somewhat overstated to me, which generally refers to dual or antagonistic behavior (e.g., minimax). I would prefer to simplify the contribution as a "mixture" (of masking modeling and generative modeling in this paper), which is a common technique in fact. Also, I would like to know how the ratios (25% - 75% in this paper) of the two strategies are determined.
6. The pipeline of using the masked pre-trained models seems still somewhat tedious, i.e., lacking in generalization. Supervised training should be performed after large-scale pre-training. Can the author provide an overall promotion compared with training from random initialization, or try zero-shot generalization on downstream tasks?

**Questions:**

1. Have you tried to pre-train separately according to different domains and then fine-tune it for domain-specific downstream tasks? As observed from Table 1, there are several discrepancies in different domains, such as the frequencies of Economics and EEG. Is it possible that separating datasets to pre-train domain-specific models works better?
2. The proposed method uses a fixed context for pre-training. Padding a large pre-training corpus, which generally contains univariate time series, into a fixed temporal/variate dimension. Will it cause a waste of computing resources?

---

> ### Author Response · Authors · 2024-11-29
> **Author Responses (1/n)**
>
> Thank you for your extensive evaluation and constructive feedback on our work. We hope the following clarifications and additional experiments adequately address the points raised.
>
> ---
> (1) ***Adaptation of the masked data modelling for time series analysis (e.g. regarding the masking ratio)***
>
> We would like to clarify that our contributions include the domain-specific tokenisation, the dual masking strategy, and the normalised cross-correlation loss, all of which are specifically designed for time series analysis. Additionally, we would like to emphasise that masked data modelling (MDM) is a widely adopted pre-training strategy in time series [6][7][8][9][10][12][21][24], primarily because it does not rely on heavy data augmentations difficult to design for sequential data [23]. Time series variates often exhibit high correlations, making higher masking ratios beneficial compared to the imaging modality, as they help eliminate redundancies in the learned representations. In our pre-training, we empirically set the masking ratio to 75%. Prior studies on MDM for time series, such as Ti-MAE [24], have explored optimal masking ratio for this modality. Their findings suggest that, similar to MDM in imaging, a masking ratio of 75% translates to best downstream performance.
>
> [6] Goswami, M. et al. "MOMENT: A Family of Open Time-series Foundation Models." International Conference on Machine Learning (ICML). 2024.
>
> [7] Woo, G. et al. "Unified Training of Universal Time Series Forecasting Transformers." International Conference on Machine Learning (ICML). 2024.
>
> [8] Yang, C. et al. “Biot: Biosignal transformer for cross-data learning in the wild.” Advances in Neural Information Processing Systems (NeurIPS). 2024.
>
> [9] Jiang, W. et al. “Large brain model for learning generic representations with tremendous EEG data in BCI.” International Conference on Learning Representations (ICLR). 2024.
>
> [10] Nie, Y. et al. “A time series is worth 64 words: Long-term forecasting with transformers.” International Conference on Learning Representations (ICLR). 2023.
>
> [12] Dong, J. et al. “SimMTM: A simple pre-training framework for masked time-series modeling.” Advances in Neural Information Processing Systems (NeurIPS). 2024.
>
> [21] Turgut, O. et al. "Unlocking the diagnostic potential of ecg through knowledge transfer from cardiac mri." arXiv preprint arXiv:2308.05764. 2023.
>
> [23] Assran, M. et al. "Self-supervised learning from images with a joint-embedding predictive architecture." Conference on Computer Vision and Pattern Recognition (CVPR). 2023.
>
> [24] Li, Z. et al. "Ti-mae: Self-supervised masked time series autoencoders." arXiv preprint arXiv:2301.08871. 2023.
>
> ---
> (2) ***Can a shared patch projector reflect different semantics among variates and domains? Could using different patch sizes for different frequencies, as in MOIRAI [7], lead to further improvements?***
>
> The authors of MOIRAI [7] (2024) presented a subsequent study [26] (2024) in which they eliminate the dependency on multiple projection layers for different frequencies. Instead, they employ a *shared* projection layer with a unified patch size across all frequencies (i.e. domains). They argue that frequencies are not a reliable indicator of the underlying patterns in time series, and that human-imposed inductive biases may hinder model generalisability. We agree with the authors and believe that projection layers should be viewed as general feature extractors, independent of frequency, variate, or domain. The extracted features serve as a learned vocabulary, which can then be slightly modulated to the domain and variate through specific positional embeddings, as implemented in OTiS.
>
> [26] Liu, X. et al. "Moirai-MoE: Empowering Time Series Foundation Models with Sparse Mixture of Experts." arXiv preprint arXiv:2410.10469. 2024.
>
> ---
> (3) ***How can domain-specific variate embeddings capture inter-variate relationships? These learned embeddings may limit  generalisability, as they do not translate to unseen variates/domains during inference.***
>
> To investigate whether adaptation to unseen domains is required for competitive performance, we have conducted additional experiments under zero-shot conditions, as detailed in Appendix F. The zero-shot results in unseen domains, such as EMG, reveal that OTiS outperforms baseline models even without domain-specific fine-tuning, underscoring the generalisability of its extracted time series features. We have included these observations in Appendix F and reworked the experiments section to present the zero-shot results.

---

> ### Author Response · Authors · 2024-11-29
> **Author Responses (2/n)**
>
> (4) ***Additional baselines and interpretation of the results***
>
> We have added TimesNet [16] and iTransformer [22] as baselines for the classification and regression tasks, respectively. Moreover, we thank the reviewer for pointing out the performance differences between the ETT\*1 and ETT\*2 (both Electricity Transformer Temperature) datasets, which we also noticed during our study. The experiments indicate that the prediction of ETT\*2 is generally easier than ETT\*1 across all baselines. For both datasets, we have analysed the distribution shapes and the frequency components. Our findings reveal that ETT\*1 exhibits long-tailed distributions and consistently includes large spikes, which may contribute to the increased difficulty in forecasting. Since the ETT\*1 and ETT\*2 were collected from two distinct regions in China [25], the external influences on the two transformers may greatly differ. For instance, one transformer may be positioned outside a steam vent or in a sunny spot, making its temperature harder to predict due to the influence of undocumented external signals.
>
> [16] Wu, H. et al. "TimesNet: Temporal 2D-Variation Modeling for General Time Series Analysis." International Conference on Learning Representations (ICLR). 2022.
>
> [22] Liu, Y. et al. "iTransformer: Inverted Transformers Are Effective for Time Series Forecasting." International Conference on Learning Representations (ICLR). 2023.
>
> [25] Zhou, H. et al. "Informer: Beyond efficient transformer for long sequence time-series forecasting." AAAI Conference on Artificial Intelligence (AAAI). 2021.
>
> ---
> (5) ***Additional ablation study on the composition of the dual masking strategy***
>
> The composition of the masking schemes is empirically set to 75% random masking and 25% post-fix masking. We have included an ablation study on the composition of the masking schemes in Appendix G.1.
>
> ---
> (6) ***Comparison with training from random initialisation and additional experiments in zero-shot settings***
>
> We have reworked the experiments section to include a randomly initialised OTiS that is trained fully supervised. The results confirm the widely reported advantages of pre-training [4][5][6][7][8][9]. Additionally, we have conducted an ablation study to investigate different pre-training strategies for OTiS on EEG event type classification, as detailed in Appendix G.2, which further stress these findings. Moreover, we have conducted experiments under zero-shot conditions. The zero-shot results in unseen domains, such as EMG, reveal that OTiS outperforms baseline models even without domain-specific training, underscoring the generalisability of its extracted time series features. We have included these observations in Appendix F and reworked the experiments section to present the zero-shot results.
>
> [4] Jin, M. et al. "Time-LLM: Time Series Forecasting by Reprogramming Large Language Models." International Conference on Learning Representations (ICLR). 2023.
>
> [5] Zhou, T. et al. "One fits all: Power general time series analysis by pretrained lm." Advances in Neural Information Processing Systems (NeurIPS). 2024.
>
> [6] Goswami, M. et al. "MOMENT: A Family of Open Time-series Foundation Models." International Conference on Machine Learning (ICML). 2024.
>
> [7] Woo, G. et al. "Unified Training of Universal Time Series Forecasting Transformers." International Conference on Machine Learning (ICML). 2024.
>
> [8] Yang, C. et al. “Biot: Biosignal transformer for cross-data learning in the wild.” Advances in Neural Information Processing Systems (NeurIPS). 2024.
>
> [9] Jiang, W. et al. “Large brain model for learning generic representations with tremendous EEG data in BCI.” International Conference on Learning Representations (ICLR). 2024.

---

> ### Author Response · Authors · 2024-11-29
> **Author Responses (3/n)**
>
> (Q1) ***Do models pre-trained on a specific domain outperform those pre-trained across domains?***
>
> We evaluate our model against several domain-specific baselines that are either i) fully supervised or ii) pre-trained and fine-tuned exclusively on the target dataset. These include N-BEATS [15], TimesNet [16], Autoformer [20], DLinear [18], MAE [21], ViT [21], iTransformer [22], CM-AE [19], MMCL [21], and PatchTST [10]. The experiments show that OTiS outperforms such domain-specific approaches in 10 out of 15 benchmarks, with inferior performance observed in only 2 out of 15 benchmarks. We have conducted additional ablation studies to investigate different pre-training strategies for OTiS in the context of EEG event type classification. The results show that domain-specific pre-training does not provide improved downstream performance compared to pre-training across domains. We have added these observations to Appendix G.2.
>
> [10] Nie, Y. et al. “A time series is worth 64 words: Long-term forecasting with transformers.” International Conference on Learning Representations (ICLR). 2023.
>
> [15] Oreshkin, B. et al. "N-BEATS: Neural basis expansion analysis for interpretable time series forecasting." International Conference on Learning Representations (ICLR). 2019.
>
> [16] Wu, H. et al. "TimesNet: Temporal 2D-Variation Modeling for General Time Series Analysis." International Conference on Learning Representations (ICLR). 2022.
>
> [18] Zeng, A. et al. "Are transformers effective for time series forecasting?" AAAI Conference on Artificial Intelligence (AAAI). 2023.
>
> [19] Radhakrishnan, A. et al. "Cross-modal autoencoder framework learns holistic representations of cardiovascular state." Nature Communications. 2023.
>
> [20] Wu, H. et al. “Autoformer: Decomposition transformers with auto-correlation for long-term series forecasting.” Advances in Neural Information Processing Systems (NeurIPS). 2021.
>
> [21] Turgut, O. et al. "Unlocking the diagnostic potential of ecg through knowledge transfer from cardiac mri." arXiv preprint arXiv:2308.05764. 2023.
>
> [22] Liu, Y. et al. "iTransformer: Inverted Transformers Are Effective for Time Series Forecasting." International Conference on Learning Representations (ICLR). 2023.
>
> ---
> (Q2) ***Does padding a large pre-training corpus to a fixed temporal/variate dimension waste computational resources?***
>
> We would like to clarify that we do not pad our pre-training corpus offline, as doing so would waste memory and limit scalability. Instead, we pad the variate dimension to the maximum number of variates *within each batch*. Furthermore, we use attention masking to ignore the padded tokens during gradient calculation, thus preventing a waste of computational resources.

---

> ### Comment · Reviewer_tXLU · 2024-12-02
>
> Thank you for the replies. I have read the rebuttal and the revision, which addressed my concerns regarding the performance and method design.  But there are a few unsolved concerns:
>
> **Regarding W2**: Although the authors provide another work to support it, I would be interested to know if the authors have done some specific empirical evaluations to draw this conclusion.
>
> **Regarding W3**: I don't think the author answered my question. I agree that learnable embeddings can help the model distinguish the heterogeneity of data in different domains (which may lead to what the rebuttal has mentioned: OTiS can outperform baseline models without domain-specific fine-tuning). However, I still cannot be convinced that the "learnable embeddings can explicitly capture inter-variate relationships" mentioned in this work.
>
> **Regarding Q1**: I read Appendix G.2 carefully: the authors provide further experiments to prove that the pre-trained model can outperform the models supervised-trained (or self-supervised trained + fine-tuned) **on one dataset**. The results are not convincing to me because of (1) the lack of state-of-the-art baseline models and self-supervised training methods, such as PatchTST, and (2) The results cannot solve the concern of OTiS in **data scaling**. Concretely, training OTiS on (1) a set of datasets (not one) related to the target dataset and (2) a domain-universal dataset (which may include the former set and some less related datasets). Is it possible that the first model that is pre-trained on a smaller scale works better?

---

> > ### Author Response · Authors · 2024-12-02
> > **Author Responses (1/n)**
> >
> > Thanks a lot for the constructive interaction! We are happy to hear that our rebuttal addressed most of your concerns. Below, we provide detailed responses to your new comments on a point-by-point basis.
> >
> > ---
> > (1) ***Can a shared patch projector reflect different semantics among variates and domains?***
> >
> > We see projection layers with a unified patch size as general feature extractors, independent of the sampling frequency, variate, or domain. This hypothesis is not derived from empirical evaluation, but based on conceptual considerations that we made prior to our study on time series foundation models, as elaborated in the following.
> >
> > The sampling frequency refers to the number of observations collected from a continuous signal per unit of time. The choice of sampling frequency depends on the goal of the analysis: some studies require low frequencies (e.g. $f=386$nHz to capture long-term economic trends spanning 60 years, only within 728 time points [31]), while others require high frequencies (e.g. $f = 44.1$kHz to capture rapid fluctuations in 10-second audio signals, resulting in 441,000 time points [32]). However, all *sampling frequencies* share the same purpose: to *ensure that the information relevant to the analysis is captured within the observation period (i.e., the time series).*
> >
> > Hence, we assume that a model will have access to all of the relevant information captured in a time series, if its context length is sufficiently long. Consequently, the context length, rather than the frequency itself, is the critical factor for model performance. Ideally, the model would analyse the entire time series to ground its prediction. However, especially for high-frequency time series, this is often infeasible with small patch sizes due to the computational complexity of attention-based models (the smaller the patch size, the more tokens need to be analysed for a specific context length).
> >
> > We hypothesise that adopting different patch sizes for different frequencies may be beneficial, not to reflect different semantics as assumed by the reviewer, but to enable sufficiently long context lengths. This also aligns with the authors of MOIRAI, who opted “for a larger patch size to handle high-frequency data, thereby lower[ing] the burden of the quadratic computation cost of attention while maintaining a long context length” [7].
> >
> > Based on this reasoning, we agree with the reviewer that it would be interesting to see how different patch sizes, and effectively different context lengths, affect the downstream performance of our model. Therefore, we will include a small empirical study in the final version of our manuscript, analysing the effect of different patch sizes.
> >
> > [7] Woo, G. et al. "Unified Training of Universal Time Series Forecasting Transformers." International Conference on Machine Learning (ICML). 2024.
> >
> > [31] McCracken, M. W. et al. "FRED-MD: A monthly database for macroeconomic research." Journal of Business & Economic Statistics. 2016.
> >
> > [32] Gemmeke, J. F. et al. "Audio set: An ontology and human-labeled dataset for audio events." IEEE international conference on acoustics, speech and signal processing (ICASSP). 2017.
> >
> > ---
> > (2) ***Do domain-specific variate embeddings capture inter-variate relationships?***
> >
> > Our model effectively learns the relationships between variates within a domain, purely from the data it has seen during training, as showcased in Figures 3, 7, 8, 9 of our manuscript.
> >
> > For example, the principal component analysis (PCA) presented in Figures 3 and 7 demonstrates that EEG-specific variate embeddings accurately capture the spatial arrangement of EEG variates, which correspond to actual electrodes placed on the scalp. In this context, the spatial arrangement represents the inter-variate relationships.
> >
> > Similarly, the PCA in Figure 8 indicates that ECG-specific variate embeddings correctly capture the spatial arrangement of ECG variates, which partially correspond to actual electrodes placed on the human body (e.g. V1-V6). In this context, the spatial arrangement again denotes the inter-variate relationships.
> >
> > Finally, the embedding similarity analysis in Figure 9 reveals that Weather-specific embeddings capture the physical relationships among the 21 climatological indicators described in Appendix E.2. In this case, these physical relationships represent the inter-variate relationships.
> >
> > If the reviewer has alternative interpretations of the term “inter-variate relationship”, we welcome further discussion.

---

> > ### Author Response · Authors · 2024-12-02
> > **Author Responses (4/n)**
> >
> > (3) ctd, ***Does OTiS pre-trained on domain-specific datasets outperform OTiS pre-trained across domains?***
> >
> > [8] Yang, C. et al. “Biot: Biosignal transformer for cross-data learning in the wild.” Advances in Neural Information Processing Systems (NeurIPS). 2024.
> >
> > [9] Jiang, W. et al. “Large brain model for learning generic representations with tremendous EEG data in BCI.” International Conference on Learning Representations (ICLR). 2024.
> >
> > [10] Nie, Y. et al. “A time series is worth 64 words: Long-term forecasting with transformers.” International Conference on Learning Representations (ICLR). 2023.
> >
> > [33] Van Dijk, H. et al. "The two decades brainclinics research archive for insights in neurophysiology (TDBrain) database." Scientific data. 2022.
> >
> > [34] Zheng, W. et al. "Investigating critical frequency bands and channels for EEG-based emotion recognition with deep neural networks." IEEE Transactions on autonomous mental development. 2015.
> >
> > [35] Obeid, Iyad, and Joseph Picone. "The temple university hospital EEG data corpus." Frontiers in neuroscience. 2016.
> >
> > [36] Song, Y. et al. "Transformer-based spatial-temporal feature learning for EEG decoding." arXiv preprint arXiv:2106.11170. 2021.
> >
> > [37] Peh, W. et al. "Transformer convolutional neural networks for automated artifact detection in scalp EEG." IEEE Engineering in Medicine & Biology Society (EMBC). 2022.
> >
> > [38] Li, H. et al. "Motor imagery EEG classification algorithm based on CNN-LSTM feature fusion network." Biomedical signal processing and control. 2022.
> >
> > [39] Jing, J. et al. "Development of expert-level classification of seizures and rhythmic and periodic patterns during eeg interpretation." Neurology. 2023.
> >
> > [40] Yang, C. et al. "Self-supervised electroencephalogram representation learning for automatic sleep staging: model development and evaluation study." JMIR AI. 2023.
> >
> > [41] Buckwalter, G. et al. "Recent advances in the TUH EEG corpus: improving the interrater agreement for artifacts and epileptiform events." IEEE Signal Processing in Medicine and Biology Symposium (SPMB). 2021.
> >
> > [42] Veloso, L. et al. "Big data resources for EEGs: Enabling deep learning research." IEEE Signal Processing in Medicine and Biology Symposium (SPMB). 2017.
> >
> > [43] Shah, V. et al. "The temple university hospital seizure detection corpus." Frontiers in neuroinformatics. 2018.
> >
> > [44] Von Weltin, E. et al. "Electroencephalographic slowing: A primary source of error in automatic seizure detection." IEEE Signal Processing in Medicine and Biology Symposium (SPMB). 2017.

---

> > ### Author Response · Authors · 2024-12-03
> > **Final Author Responses**
> >
> > Dear Reviewer ***tXLU***,
> >
> > Thanks again for engaging in a discussion.
> >
> > As you acknowledged in your response, our rebuttal has addressed your main concerns regarding performance and methodology. Additionally, we have worked to address your remaining concerns, by (1) providing a chain of thought on shared patch projectors, (2) elaborating on the terminology behind inter-variate relationships, and (3) clarifying the effectiveness of pre-training strategies, including the introduction of PatchTST [10] as a new baseline.
> >
> > We hope these efforts adequately address the remaining points you raised. If you believe so, we would greatly appreciate a final adjustment of your scores to reflect this.
> >
> > Authors
> >
> > ---
> > [10] Nie, Y. et al. “A time series is worth 64 words: Long-term forecasting with transformers.” International Conference on Learning Representations (ICLR). 2023.

---

> ### Author Response · Authors · 2024-12-02
> **Author Responses (2/n)**
>
> (3) ***Does OTiS pre-trained on domain-specific datasets outperform OTiS pre-trained across domains?***
>
> Thank you for pointing this out out; we believe there may be a misunderstanding here. Note that OTiS-Base$_\text{EEG}$ in Table 12 refers to OTiS pre-trained on TDBrain [33] and SEED [34], i.e. a set of two EEG datasets related to the target TUEV dataset [35]. In contrast, OTiS-Base refers to OTiS pre-trained on the full pre-training corpus detailed in Table 1 of our manuscript.
>
> The additional experiments on the TUEV [35] data provided in Appendix G.2 show that both OTiS-Base$_\text{EEG}$ and OTiS-Base outperform i) domain-specific baselines (either fully supervised or pre-trained and fine-tuned on the target dataset, i.e. one dataset), ii) general baselines (pre-trained on few external source datasets and fine-tuned on the target dataset), and even iii) foundation models (pre-trained on multiple external source datasets and fine-tuned on the target dataset). The domain-specific baselines include ST-Transformer [36], CNN-Transformer [37], FFCL [38], and SPaRCNet [39]. The general methods include ContraWR [40]. The foundation methods include BIOT [8] and LaBraM [9]. We would like to clarify that other than stated by the reviewer, these latter models represent state-of-the-art baselines that are trained using self-supervised learning. Additionally, we have introduced PatchTST [10] as a new baseline. We have reworked Appendix G.2 to summarise the baselines, similar as in the following Table.
>
> | **Model**      	| **Pre-training Method**   | **Pre-training Dataset**  | **Domain Adaptation** | **Architecture**    	|
> |--------------------|:--------------------:|:-----------:|:------------------------:|-------------------------|
> | ST-Transformer [36] 	| $-$             	| Target	| Fine-tuning        	| Transformer         	|
> | CNN-Transformer [37]	| $-$             	| Target	| Fine-tuning        	| CNN and Transformer 	|
> | FFCL [38]          	| $-$             	| Target	| Fine-tuning        	| CNN and LSTM        	|
> | SPaRCNet [39]      	| $-$             	| Target	| Fine-tuning        	| 1D-CNN              	|
> | ContraWR [40]      	| CL             	| Target     	| Fine-tuning        	| Transformer         	|
> | PatchTST [10]      	| MDM             	| Target     	| Fine-tuning        	| Transformer         	|
> | BIOT [8]         	| MDM            	| $*$     	| Fine-tuning        	| Transformer         	|
> | LaBraM [9]        	| MDM            	| $^+$    	| Fine-tuning        	| Transformer         	|
>
> $*$ Pre-trained on 6 EEG datasets (totalling 13,000 recording hours), including the target dataset (i.e. TUEV [35])
>
> $^+$ Pre-trained on 16 EEG datasets (totalling 2,500 recording hours), including TUAR [41], TUEP [42], TUSZ [43], and TUSL [44], which are subsets of the TUH [35]. As TUEV [35] is a subset of TUH [35], too, there may be potential information leakage through overlapping subjects between subsets.

---

> ### Author Response · Authors · 2024-12-02
> **Author Responses (3/n)**
>
> (3) ctd, ***Does OTiS pre-trained on domain-specific datasets outperform OTiS pre-trained across domains?***
>
> We have updated Table 12 accordingly with the results of PatchTST [10], as outlined in the following.
>
> | **Methods**                         	| **Parameters** | **Balanced ACC** ⬆️   	| **Cohen’s Kappa** ⬆️   	| **Weighted F1** ⬆️      	|
> |-----------------------------------------|----------------|---------------------------|----------------------------|-----------------------------|
> | ST-Transformer [36]               	| 3.5M       	| 0.3984 ± 0.0228      	| 0.3765 ± 0.0306       	| 0.6823 ± 0.0190        	|
> | CNN-Transformer [37]              	| 3.2M       	| 0.4087 ± 0.0161      	| 0.3815 ± 0.0134       	| 0.6854 ± 0.0293        	|
> | FFCL [38]                         	| 2.4M       	| 0.3979 ± 0.0104      	| 0.3732 ± 0.0188       	| 0.6783 ± 0.0120        	|
> | SPaRCNet [39]                     	| 0.79M      	| 0.4161 ± 0.0262      	| 0.4233 ± 0.0181       	| 0.7024 ± 0.0104        	|
> | ContraWR [40]                     	| 1.6M       	| 0.4384 ± 0.0349      	| 0.3912 ± 0.0237       	| 0.6893 ± 0.0136        	|
> | PatchTST [10]                     	| 3.3M       	| 0.4677 ± 0.0243      	| 0.5051 ± 0.0169       	| 0.7526 ± 0.0203        	|
> | BIOT  [8]             	| 3.2M       	| 0.5281 ± 0.0225      	| 0.5273 ± 0.0249       	| 0.7492 ± 0.0082        	|
> | LaBraM [9]           	| 369M       	| **0.6616 ± 0.0170**  	| **0.6745 ± 0.0195**   	| **0.8329 ± 0.0086**    	|
> |                                     	|            	|                       	|                        	|                         	|
> | OTiS-Base$_\text{w/o pre-training}$*	| 8M         	| 0.5361 ± 0.0350      	| 0.5183 ± 0.0316       	| 0.7642 ± 0.0157        	|
> | OTiS-Base$_\text{EEG}$$^\dagger$    	| 8M         	| 0.5562 ± 0.0106      	| 0.5504 ± 0.0204       	| 0.7784 ± 0.0095        	|
> | OTiS-Base                           	| 8M         	| _0.5743 ± 0.0257_    	| _0.5913 ± 0.0146_     	| _0.8004 ± 0.0071_      	|
> |                                     	|            	|                       	|                        	|                         	|
>
> $*$ Model was randomly initialized and trained fully supervised.
>
> $^\dagger$ Model was pre-trained only with the EEG data of our pre-training corpus (i.e. TDBrain [33] and SEED [34]).
>
> The experiments reveal that general models (i.e., OTiS-Base$_\text{EEG}$), trained on a smaller scale, do not perform better than foundation models (i.e., OTiS-Base and LaBraM), trained on a large scale.
>
> We have revised Appendix G.2 to more carefully highlight these observations and hope that our discussion and clarifications adequately address the points you raised. If there are any remaining questions or concerns, we are happy to discuss them further.

---

### Official Review · Reviewer_yUJh · 2024-10-31

**Soundness:** 2
**Presentation:** 3
**Contribution:** 2
**Rating:** 5
**Confidence:** 4

**Summary:**

This paper presents OTiS for multi-domain time series analysis, building on existing pre-training paradigms for time series. It allocates domain-specific variable embeddings to distinguish the heterogeneity of different variables across domains and enhances the model's ability to learn temporal causal relationships through a dual-masking strategy. Additionally, it introduces NCC loss to capture global patterns. Experimental results demonstrate that the proposed method achieves competitive performance in time series classification, regression, and forecasting tasks across multiple domains compared to SOTA methods. Visualization results further highlight the effectiveness and interpretability of the domain-specific variable embeddings.

**Strengths:**

1. The paper is well-written, and the method is easy to understand. The authors clearly articulate how they consider the heterogeneity of different domain time series to achieve multi-domain time series forecasting.

2. This paper focuses on the problem of multi-domain time series analysis, which is crucial for building generalizable foundational models for time series.

3. The experimental section utilizes a large amount of data, and the model is open-source, contributing particular engineering value to the community.

**Weaknesses:**

1. The paper mentions that one of the challenges of cross-domain time series models is the significant differences in temporal dynamics and sampling frequencies among different domains. However, the paper uses the same patch size for all domains when dividing patches, failing to accommodate the unique sampling rates of different domains. This oversight means the paper does not sufficiently consider the differences in sampling rates across domains. Additionally, using a shared patch projector to encode the temporal dynamics within each patch does not adequately address the differences in temporal dynamics between domains. While this approach may be common in previous works, it does not consider the temporal heterogeneity among domains.

2. The method of considering variable heterogeneity through learned variable embeddings is not uncommon. In spatiotemporal prediction, some methods [2][3] have already employed learnable embeddings to explicitly distinguish heterogeneous spatiotemporal patterns by learning time-specific and space-specific parameter spaces.

3. [1] proposed using textual descriptions to label different time series domains for cross-domain time series forecasting, utilizing a channel-independent strategy. In contrast, the domain-specific variable embeddings in this paper correspond to a channel-mixing strategy. I look forward to seeing a comparison between these two strategies in cross-domain time series.

4. The experimental section lacks details about the baselines. How were these methods selected? Were they pre-trained and fine-tuned? If so, what data was used for pre-training and fine-tuning?

5. How does the performance of the proposed method compare to conventional time series classification or forecasting methods trained on a single specific dataset?

[1] UniTime: A Language-Empowered Unified Model for Cross-Domain Time Series Forecasting, WWW, 2024

[2] Heterogeneity-Informed Meta-Parameter Learning for Spatiotemporal Time Series Forecasting, KDD, 2024

[3] Adaptive Graph Convolutional Recurrent Network for Traffic Forecasting, NeurIPS, 2020

**Questions:**

See the weaknesses

---

> ### Author Response · Authors · 2024-11-29
> **Author Responses (1/n)**
>
> Thank you for your engaging comments and the thorough evaluation. We hope the following clarifications and additional experiments adequately address the points raised.
>
> ---
> (1) ***A patch projector with a unified patch size, shared across all domains, fails to account for unique sampling frequencies and does not adequately address differences in temporal dynamics***
>
> Recent state-of-the-art foundation models, such as MOIRAI [7] (2024), introduce multiple projection layers with different patch sizes to handle distinct frequencies. However, the authors of MOIRAI presented a subsequent study [26] (2024) in which they eliminate the dependency on multiple projection layers for different frequencies. Instead, they employ a *shared* projection layer with a unified patch size across all frequencies (i.e. domains). They argue that frequencies are not a reliable indicator of the underlying patterns in time series, and that human-imposed inductive biases may hinder model generalisability. We agree with the authors and believe that projection layers should be viewed as general feature extractors, independent of frequency, variate, or domain. The extracted features serve as a learned vocabulary, which can then be slightly modulated to the domain and variate through specific positional embeddings, as implemented in OTiS.
>
> [7] Woo, G. et al. "Unified Training of Universal Time Series Forecasting Transformers." International Conference on Machine Learning (ICML). 2024.
>
> [26] Liu, X. et al. "Moirai-MoE: Empowering Time Series Foundation Models with Sparse Mixture of Experts." arXiv preprint arXiv:2410.10469. 2024.
>
> ---
> (2) ***Considering variate heterogeneity through learned embeddings is not uncommon***
>
> Prior to the stated works [2][3], learnable 2D positional embeddings were extensively studied by Dosovitskiy et al. [29], and we do not claim this as a novel aspect of our study. Instead, we introduce a unique approach by employing non-learnable temporal embeddings (shared across domains) and learnable variate embeddings (specific to each domain). This special composition of the “2D” positional embeddings represents a novel contribution of our study, which to the best of our knowledge has not been explored in time series analysis before.
>
> [2] Heterogeneity-Informed Meta-Parameter Learning for Spatiotemporal Time Series Forecasting, KDD, 2024
>
> [3] Adaptive Graph Convolutional Recurrent Network for Traffic Forecasting, NeurIPS, 2020
>
> [29] Dosovitskiy, A. et al. "An image is worth 16x16 words: Transformers for image recognition at scale." Advances in Neural Information Processing Systems (NeurIPS). 2020.
>
> ---
> (3) ***Comparison with channel-independent baselines***
>
> We have conducted extensive benchmarking to compare our model against several state-of-the-art baselines employing *channel-independent* strategies, similar to the proposed UniTime [1]. These include N-BEATS [15], TimesNet [16], TF-C [17], DLinear [18], PatchTST[10], CM-AE [19], Time-LLM [4], GPT4TS [5], and MOMENT [6]. Covering all key use cases in time series analysis, such as classification, regression, and forecasting, these baselines provide a comprehensive comparison. Our experiments reveal that OTiS outperforms such channel-independent approaches in 10 out of 15 benchmarks, with inferior performance to such approaches in only 2 out of 15 benchmarks. These results validate the effectiveness of OTiS’ channel-mixing strategy.
>
> [1] UniTime: A Language-Empowered Unified Model for Cross-Domain Time Series Forecasting, WWW, 2024
>
> [4] Jin, M. et al. "Time-LLM: Time Series Forecasting by Reprogramming Large Language Models." International Conference on Learning Representations (ICLR). 2023.
>
> [5] Zhou, T. et al. "One fits all: Power general time series analysis by pretrained lm." Advances in Neural Information Processing Systems (NeurIPS). 2024.
>
> [6] Goswami, M. et al. "MOMENT: A Family of Open Time-series Foundation Models." International Conference on Machine Learning (ICML). 2024.
>
> [10] Nie, Y. et al. “A time series is worth 64 words: Long-term forecasting with transformers.” International Conference on Learning Representations (ICLR). 2023.
>
> [15] Oreshkin, B. et al. "N-BEATS: Neural basis expansion analysis for interpretable time series forecasting." International Conference on Learning Representations (ICLR). 2019.
>
> [16] Wu, H. et al. "TimesNet: Temporal 2D-Variation Modeling for General Time Series Analysis." International Conference on Learning Representations (ICLR). 2022.
>
> [17] Zhang, X. et al. "Self-supervised contrastive pre-training for time series via time-frequency consistency." Advances in Neural Information Processing Systems (NeurIPS). 2022.
>
> [18] Zeng, A. et al. "Are transformers effective for time series forecasting?" AAAI Conference on Artificial Intelligence (AAAI). 2023.
>
> [19] Radhakrishnan, A. et al. "Cross-modal autoencoder framework learns holistic representations of cardiovascular state." Nature Communications. 2023.

---

> > ### Comment · Reviewer_yUJh · 2024-12-02
> >
> > Thank you for the response. I still have some concerns. The response mentions that the innovative contribution of the paper lies in the special combination of using non-learnable domain-shared embeddings in the time dimension and domain-specific learnable variable embeddings in the variable dimension. However, it seems that there is a lack of empirical evaluation of this design, so it remains unclear whether this approach is truly effective. Furthermore, the revised version still appears to lack a comparison with UniTS, which is a recent method for cross-domain time series modeling that distinguishes different domains through text representations. A comparison with UniTS would help verify the effectiveness of using domain-specific variable embeddings in this paper.

---

> ### Author Response · Authors · 2024-11-29
> **Author Responses (2/n)**
>
> (4) ***Clarification of the baseline models***
>
> We have reworked the experiments section to clarify the categorisation of the baselines. Additionally, we have included a summary of all baselines, detailing their architectures, pre-training strategies, and domain adaptation techniques, in Appendix B.
>
> ---
> (5) ***Comparison with traditional baselines trained on a single, specific dataset***
>
>  In extensive benchmarking, we compare our model against multiple domain-specific baselines that are either i) fully supervised or ii) pre-trained and fine-tuned exclusively on the target dataset. These include N-BEATS [15], TimesNet [16], Autoformer [20], DLinear [18], MAE [21], ViT [21], iTransformer [22], CM-AE [19], MMCL [21], and PatchTST [10]. These baselines span all key use cases in time series analysis, providing a comprehensive comparison. The experiments show that OTiS outperforms such domain-specific approaches in 10 out of 15 benchmarks, with inferior performance to such approaches in only 2 out of 15 benchmarks. We have conducted an additional ablation study to investigate different pre-training strategies for OTiS, as detailed in Appendix G.2, which further stress the widely reported advantages of general pre-training across domains [4][5][6][7][8][9].
>
> [4] Jin, M. et al. "Time-LLM: Time Series Forecasting by Reprogramming Large Language Models." International Conference on Learning Representations (ICLR). 2023.
>
> [5] Zhou, T. et al. "One fits all: Power general time series analysis by pretrained lm." Advances in Neural Information Processing Systems (NeurIPS). 2024.
>
> [6] Goswami, M. et al. "MOMENT: A Family of Open Time-series Foundation Models." International Conference on Machine Learning (ICML). 2024.
>
> [7] Woo, G. et al. "Unified Training of Universal Time Series Forecasting Transformers." International Conference on Machine Learning (ICML). 2024.
>
> [8] Yang, C. et al. “Biot: Biosignal transformer for cross-data learning in the wild.” Advances in Neural Information Processing Systems (NeurIPS). 2024.
>
> [9] Jiang, W. et al. “Large brain model for learning generic representations with tremendous EEG data in BCI.” International Conference on Learning Representations (ICLR). 2024.
>
> [10] Nie, Y. et al. “A time series is worth 64 words: Long-term forecasting with transformers.” International Conference on Learning Representations (ICLR). 2023.
>
> [15] Oreshkin, B. et al. "N-BEATS: Neural basis expansion analysis for interpretable time series forecasting." International Conference on Learning Representations (ICLR). 2019.
>
> [16] Wu, H. et al. "TimesNet: Temporal 2D-Variation Modeling for General Time Series Analysis." International Conference on Learning Representations (ICLR). 2022.
>
> [18] Zeng, A. et al. "Are transformers effective for time series forecasting?" AAAI Conference on Artificial Intelligence (AAAI). 2023.
>
> [21] Turgut, O. et al. "Unlocking the diagnostic potential of ecg through knowledge transfer from cardiac mri." arXiv preprint arXiv:2308.05764. 2023.
>
> [22] Liu, Y. et al. "iTransformer: Inverted Transformers Are Effective for Time Series Forecasting." International Conference on Learning Representations (ICLR). 2023.

---

> ### Author Response · Authors · 2024-12-02
> **Author Responses**
>
> Thanks a lot for the quick response and the suggestion! We would like to clarify that the suggested UniTS [30] does not utilise any text representations. Instead, it employs learnable *domain-agnostic* variate embeddings (i.e., learnable embeddings shared across all domains) to implicitly accommodate distinct domains.
>
> To empirically evaluate the effectiveness of our learnable *domain-specific* variate embeddings, we have conducted an ablation study, as described in the experiments section. In this study, we replaced the learnable domain-specific embeddings with learnable domain-agnostic variate embeddings, similar to the approach in UniTS [30]. The results, presented in Figure 5, highlight two key advantages of domain-specific variate embeddings: (i) enhanced robustness, evidenced by a smaller interquartile range, and (ii) improved downstream performance.
>
> Additionally, in extensive benchmarking experiments we compare our model against several state-of-the-art baselines, including those that leverage text representations to distinguish between domains. For instance, in Time-LLM [4], the authors encode explicit descriptions of both the dataset and the domain as text representations, which are then used with the time series representations to perform forecasting tasks. Our experiments demonstrate that OTiS outperforms such approaches in 4 out of 6 forecasting benchmarks, effectively validating the utility of learnable domain-specific variate embeddings.
>
> We have updated the related works section to include a discussion of UniTS [30] and hope that these experiments and comparisons adequately address the point you raised. If there are any remaining questions or concerns, we would be happy to discuss them further.
>
> ---
> [4] Jin, M. et al. "Time-LLM: Time Series Forecasting by Reprogramming Large Language Models." International Conference on Learning Representations (ICLR). 2023.
>
> [30] Gao, S. et al. "UniTS: A unified multi-task time series model." Advances in Neural Information Processing Systems (NeurIPS). 2024.

---

> ### Author Response · Authors · 2024-12-03
> **Final Author Responses**
>
> Dear Reviewer ***yUJh***,
>
> Thank you once again for engaging in a discussion.
>
> In your previous response, you raised the concern whether the domain-specific variate embeddings used in our study are effective in distnguishing different domains. You suggested comparing our model against the UniTS [30] model, which uses domain-agnostic embeddings, and baselines that use textual representations to differentiate between domains. To address this concern, we have (1) conducted an ablation study to investigate domain-agnostic embeddings, similar to UniTS [30], and (2) compared our model against the Time-LLM [4] model, which uses textual representations for domain differentiation. These experiments demonstrate that our domain-specific variate embeddings are most effective in distnguishing different domains, yielding to robust improvements in downstream performance.
>
> We hope these efforts adequately address the remaining point you raised. If you believe so, we would greatly appreciate a final adjustment of your scores to reflect this.
>
> Authors
>
> ---
> [4] Jin, M. et al. "Time-LLM: Time Series Forecasting by Reprogramming Large Language Models." International Conference on Learning Representations (ICLR). 2023.
>
> [30] Gao, S. et al. "UniTS: A unified multi-task time series model." Advances in Neural Information Processing Systems (NeurIPS). 2024.

---

### Official Review · Reviewer_UJog · 2024-11-01

**Soundness:** 2
**Presentation:** 2
**Contribution:** 2
**Rating:** 5
**Confidence:** 3

**Summary:**

Authors spot the important fact that the variate structure is heterogeneous across domains and this structure may represent more complex relationships. Thus, they propose a time series pre-training pipeline called OTiS. The OTiS is composed of a specially designed tokenizer that can add domain-specific signature to the time series and a novel loss for pretraining.

**Strengths:**

- The paper is well-written and easy to follow.

- Authors spot the important fact that the variate structure is heterogeneous across domains and this structure may represent more complex relationships.

- The visualization for variate embedding seems to be interesting and insightful.

- A substantial portion of this research focuses on EEG signals, which presents a novel and promising approach. The authors introduce an innovative method to model a "specific set of systems" that, despite being observed differently—such as TDBrain and SEED with 19 channels versus LEMON with 62 channels—remain comparable.

**Weaknesses:**

- As noted in the strengths, this work addresses the challenge of generalizing across datasets that contain time series of similar systems but are recorded differently, such as variations in sampling rates and physical values. However, the claims regarding cross-domain generalization may be overstated.

- From the perspective of generalized time series analysis, the primary contribution of variate-specific embedding may not be effective in other systems where the interrelationships between variates are not as straightforward as their spatial arrangement (e.g., the electrodes in EEG as depicted in Figure 3 of the manuscript). In different physical systems, two variates may exhibit complex computational relationships (e.g., voltage and current as described by Ohm's Law), complicating the direct modeling of variates as embeddings.

**Questions:**

- How does the domain-specific tokenizer adapt to unseen domains with distinct variate structures?

- Additionally, how does the domain-specific tokenizer generalize across different systems within the same domain? For instance, while both electrical transformers and power generators belong to the "energy" domain, they exhibit differing properties and produce distinct time series readings. How does the sub-domain adaptation discussed in Section 3.1 address this scenario?

- A broader question, not specific to this paper: At what level of granularity should we define the domain?

**Details Of Ethics Concerns:**

Authors use Github Link to share the code, leading to potential personal information leakage. This may require further investigation.

---

> ### Author Response · Authors · 2024-11-29
> **Author Responses (1/n)**
>
> Thank you for your thoughtful comments on our work. We hope the following clarifications and additional analyses adequately address the points raised.
>
> ---
> (1) ***Claims regarding the generalisation across domains may be overstated***
>
> See (Q1)
>
> ---
> (2) ***Analysis of variate embeddings in domains with complex relationships (i.e. where variates are not as straightforward as their spatial arrangement)***
>
> We have analysed further domain-specific variate embeddings in Appendix E.2, showcasing that OTiS is capable of capturing complex inter-variate relationships. We acknowledge the concerns of the reviewer that high correlations between spatially proximate variates (e.g. EEG electrodes) might facilitate learning these relationships. The reviewer has suggested investigating the relationship between voltage $U$ [V] and current $I$ [A], described by $U = R * I$, where $R$ [Ohm] denotes resistance. However, the *linear* relationship between these two variates may represent a trivial case, while scenarios involving more complex (i.e. *non-linear*) inter-variate relationships would offer deeper insight into OTiS’ modelling capabilities. To this end, the Weather dataset [14] provides a more suitable test case, spanning diverse climatological categories such as temperature, humidity, wind, radiation, pressure, and precipitation, which exhibit non-linear relationships. As detailed in Appendix E.2, our exploration of Weather-specific variate embeddings learned during fine-tuning demonstrates that OTiS effectively models such complex relationships.
>
> [14] Max Planck Institute for Biogeochemistry. “Weather station.” 2024. https://www.bgc-jena.mpg.de/wetter/.
>
> ---
> (Q1) ***How does the domain-specific tokeniser adapt to unseen domains?***
>
> Let $S$ denote a previously unseen domain with $V_S$ variates and $D$ denote the embedding dimension of our model. We randomly initialise variate embedding $E_S^V \in R^{V_S \times D}$ and fine-tune them along with the encoder and, if required, the decoder, for the specific application in $S$. To investigate whether adaptation to unseen domains is even necessary for competitive performance, we have conducted additional experiments under zero-shot conditions, as detailed in Appendix F. The zero-shot results in unseen domains, such as EMG, reveal that OTiS outperforms baseline models even without domain-specific fine-tuning, underscoring the generalisability of its extracted time series features. We have included these observations in Appendix F and reworked the experiments section to present the zero-shot results.
>
> ---
> (Q2) ***How does the domain-specific tokeniser generalise across different systems within the same domain (e.g. electrical transformers and power generators in an imaginary "Energy" domain)?***
>
> Similar to how we separate the EEG and ECG domains in our pre-training corpus, rather than combining them under a broader “Medicine” domain, one could similarly define distinct domains for electrical transformers and power generators. See (Q3) for a more detailed discussion on the definition of domains.
>
> ---
> (Q3) ***At what level of granularity should domains be defined?***
>
> In general, the level of granularity at which to define a domain depends on the underlying characteristics of the data. We believe that a domain should be defined at a level where the data shares meaningful patterns, particularly with respect to inter-variate relationships and temporal dynamics. For example, we define the NN5 dataset [27] (daily cash withdrawals) as ‘Banking’ domain and the FRED-MD dataset [28] (macro-economic indicators) as ‘Economics’ domain, even though both could broadly fall under a ‘Finance’ domain. However, the Banking domain is characterised by high periodicity and little long-term trends, whereas the Economics domain exhibits the opposite. The key is to balance between a too broad definition, which may obscure important patterns, and too narrow definition, which may limit generalisation. As discussed in our limitations section, automated pipelines that leverage embedding similarities to compare datasets could aid in defining domains, reducing reliance on human-imposed inductive biases.
>
> [27] Taieb, S. et al. "A review and comparison of strategies for multi-step ahead time series forecasting based on the NN5 forecasting competition." Expert systems with applications. 2012.
>
> [28] McCracken, M. W. et al. "FRED-MD: A monthly database for macroeconomic research." Journal of Business & Economic Statistics. 2016.
>
> ---
> (Ethical concerns) ***Authors use a GitHub link to share the code, which could lead to personal information leakage and may require further investigation.***
>
> Regarding the ethics concerns, we have carefully set up the GitHub repository upon submission, excluding any identifying metadata, commit histories, or personal information. We thus strictly adhere to anonymity guidelines while maintaining reproducibility.

---

> > ### Comment · Reviewer_UJog · 2024-12-03
> >
> > Sorry for the late reply and welcome back to the party.
> >
> > I appreciate authors' effort to address my concern and most of my concerns are adequately addressed. After checking the Appendix E.2 and Appendix F, now I think this is a work with insights about extracting correlation in channels. Thus, I will bump up my score.
> >
> > I still have concerns about whether the large scale pre-training paradigm of time series forecasting works as some researchers claims. Since the model may just memorizing the patterns and some "foundation models" even fail to predict the sine wave correctly, e.g. moirai. However, this may not in the scale of authors' research. So, I will lower my confidence to scale down the weight of my score.

---

> > > ### Author Response · Authors · 2024-12-03
> > > **Author Responses**
> > >
> > > Thanks a lot for your response, it is great to have you back!
> > >
> > > Inspired by your comment, we have prepared a small experiment, which you can find in the README of our anonymous GitHub repository (https://github.com/OTiS-official/OTiS).
> > >
> > > Spoiler alert: ***Contrary to the assumption that our model only learns correlations across variates, new experiments reveal that OTiS also captures the inherent patterns in time series, which generalise well to unseen data.***
> > >
> > > We conducted novel forecasting experiments on uni-variate sine waves with distinct frequencies, ranging from 2Hz to 100Hz. In this uni-variate setting, we ensure that our model does not leverage correlations from other variates. We have employed minimal training for these experiments: we freeze the pre-trained OTiS and train only the randomly initialised domain-specific variate embedding (a single embedding for uni-variate sine waves, totalling less than 0.2k trainable parameters). We solely train on uni-variate 50Hz sine waves. Then, during inference, we perform zero-shot forecasting on unseen uni-variate sine waves with frequencies including 2Hz, 28Hz, 60Hz, and 100Hz, using the sine-specific variate embedding learned on 50Hz sine waves. The results reveal that OTiS is not only capable of capturing inter-variate relationships (i.e. correlations across variates, as described in Appendix E.2), but also temporal dynamics and patterns of time series, which generalise to unseen data. We have updated our manuscript to include these new findings in Appendix E.
> > >
> > > Thanks again for the very useful hint regarding the sine wave experiments. Your input definitely contributed to a deeper understanding of our model’s capabilities!

---

### Official Review · Reviewer_pz7i · 2024-11-02

**Soundness:** 2
**Presentation:** 4
**Contribution:** 2
**Rating:** 5
**Confidence:** 4

**Summary:**

This paper proposes a time series model architecture and a pre-training objective. The key idea is that their architecture acknowledges that training and testing time series may have different sampling rates and variables. The authors propose a straightforward tokenization scheme to learn embeddings for different variables, which can get added onto regular patch and temporal embeddings, thereby conditioning the predictions on the measured variables. They then pre-train their model on a collection of existing datasets, and evaluate its performance by finetuning on new datasets for some forecasting, regression, and classification datasets. They find that finetuning their model on new datasets can outperform other recent methods.

**Strengths:**

This paper has many strengths:
* The key idea to condition the model on different variables and domains is good. Indeed many related works effectively ignore this information.
* The paper is overall written quite well and arguments are presented clearly.
* The experiments investigate multiple axes, including ablation of their method and different dataset and model sizes, and visualizations of the embeddings.
* The public data and model weights will help the community build on this work.

**Weaknesses:**

This paper has weaknesses to address:
* The major weakness of this paper is the extremely limited experiments section. There are many experiments, yet almost no explanation of how they're run or interpretation of the results. Most of the results are written like an advertisement, mostly just stating the method outperforms others. This leaves the reader unclear why the performance gains happen. Ultimately it's not clear when/why the findings would generalize. The result is that some claims appear to be quite overstated. For example, L423-L424 states *"embeddings of domains with shared high-level semantics cluster together, as depicted in Appendix E.1. For example, embeddings of mono and stereo audio group closely, as do those of banking and economics."* But this is cherry-picked---Temperature is way closer to Mono and Stereo Audio than Banking is to Economics.
* Similarly, many important experimental details are missing or relegated to the Appendix, and the Appendix also includes almost no explanations or interpretations. For example, the PCA experiments in Figures 3, 7, and 8 aren't explained.
* It's unclear how many variables actually overlap between training/testing, which seems to be a key element to make the model outperform others. Yet this isn't analyzed. Showing that others fail by ignoring other variables should be a key element of the experiments.

**Questions:**

Please feel free to address any misunderstandings I've stated in the weaknesses. Answers to the following questions would help me better calibrate my score:
1. How long does it take to finetune on new tasks?
2. How does the finetuned model perform compared to task-specific models? Are these testing datasets really good cases that need pre-trained models?
3. How do you get the ground truth embeddings in Figure 3?
4. Is there any intuition around what information could be shared across such different domains to make pre-training on them useful?

---

> ### Author Response · Authors · 2024-11-29
> **Author Responses (1/n)**
>
> Thank you for your careful and thorough evaluation of our work. We hope the following clarifications and additional experiments adequately address the points raised.
>
> ---
> (1) ***Explanation and interpretation of the results concerning generalisability and performance gains***
>
> We have reworked the experiments section and added discussions to Appendix F and G, explaining the results of our study in more detail. To summarise, we observe that domain-specific models (either fully supervised or pre-trained and fine-tuned exclusively on the target data) are inferior to general models (pre-trained on external source data and fine-tuned on the target data) and foundational models (pre-trained on large corpora and fine-tuned on the target data), as discussed in the experiments section and Appendix G.2. Moreover, we have conducted additional zero-shot experiments, demonstrating that our model is able to extract distinct representations for different inputs, as discussed in Appendix F. These distinct representations can be observed across domains and tasks, suggesting that the time series features extracted by OTiS are generalisable. Our experiments further show that adaptation to the specific task through fine-tuning generally boosts downstream performance.
>
> ---
> (2) ***Clarification of the principal component analysis***
>
> We have reworked the results section to clarify the principal component analysis. In particular, we have added a detailed explanation on the alignment of the EEG-specific variate embeddings with the 3D electrode coordinates of the international 10-20 system for EEG recordings to Appendix E.2. See (Q3).
>
> ---
> (3) ***Analysis of the overlap between training and testing variables***
>
> We are not entirely sure what the reviewer means by *variables*, but we have extensively analysed the effects of domain-specific variate embeddings in full training and zero-shot settings in the experiments section and the Appendix. We believe this analysis shows why our method outperforms the baselines, but if the reviewer has a specific aspect that they would like to discuss further in detail, we would happily engage.

---

> ### Author Response · Authors · 2024-11-29
> **Author Responses (2/n)**
>
> (Q1) ***How long does it take to fine-tune on new tasks?***
>
> We fine-tune our model on all tasks using a single NVIDIA RTX A6000-48GB GPU and 32 CPUs. With this setup, the training times for the three example tasks are as follows.
> | Task                     | # Steps | Training Time (s) |
> |--------------------------|---------|--------------------|
> | Classification (Epilepsy) | 150     | 90                |
> | Regression (LVSV)         | 3350    | 8400              |
> | Forecasting (ETTm2)       | 1000    | 600               |
>
> ---
> (Q2) ***How does the fine-tuned model perform compared to task-specific models? Are the test datasets representative of cases that require pre-trained models?***
>
> We evaluate our model against current state-of-the-art baselines using the established benchmark datasets in time series analysis. Our experiments confirm the widely reported advantages of pre-training for these very benchmarks [4][5][6][7][8][9][10][11][12], demonstrating that (i) general models (pre-trained on external source data and fine-tuned on the target data) and (ii) foundational models (pre-trained on large corpora and fine-tuned on the target data) outperform (iii) domain-specific models (either fully supervised or pre-trained and fine-tuned exclusively on the target data). The baselines in our experiments include 11 domain-specific models, 6 general models, and 4 foundation models. We have conducted an additional ablation study (further 4 domain-specific models, 1 general model, and 2 foundation models) to investigate different pre-training strategies for OTiS, as detailed in Appendix G.2, which further stress these findings. In conclusion, we have reworked the experiments sections to provide details on the baselines and to highlight the advantages of pre-training.
>
> [4] Jin, M. et al. "Time-LLM: Time Series Forecasting by Reprogramming Large Language Models." International Conference on Learning Representations (ICLR). 2023.
>
> [5] Zhou, T. et al. "One fits all: Power general time series analysis by pretrained lm." Advances in Neural Information Processing Systems (NeurIPS). 2024.
>
> [6] Goswami, M. et al. "MOMENT: A Family of Open Time-series Foundation Models." International Conference on Machine Learning (ICML). 2024.
>
> [7] Woo, G. et al. "Unified Training of Universal Time Series Forecasting Transformers." International Conference on Machine Learning (ICML). 2024.
>
> [8] Yang, C. et al. “Biot: Biosignal transformer for cross-data learning in the wild.” Advances in Neural Information Processing Systems (NeurIPS). 2024.
>
> [9] Jiang, W. et al. “Large brain model for learning generic representations with tremendous EEG data in BCI.” International Conference on Learning Representations (ICLR). 2024.
>
> [10] Nie, Y. et al. “A time series is worth 64 words: Long-term forecasting with transformers.” International Conference on Learning Representations (ICLR). 2023.
>
> [11] Zhang, X. et al. “Self-supervised contrastive pre-training for time series via time-frequency consistency.” Advances in Neural Information Processing Systems (NeurIPS). 2022.
>
> [12] Dong, J. et al. “SimMTM: A simple pre-training framework for masked time-series modeling.” Advances in Neural Information Processing Systems (NeurIPS). 2024.
>
> ---
> (Q3) ***How is the ground truth obtained in Figure 3?***
>
> We have added a detailed explanation of the alignment of the learned EEG-specific variate embeddings with the true electrode layout to Appendix E.2. Note that the electrode placement of all EEG datasets used in our study follows the international 10-20 system for EEG recordings [13]. However, we would like to clarify that the 3D electrode coordinates of the 10-20 EEG system are not used for training. Instead, our model implicitly learns to model the spatial structure solely from the EEG recordings seen during training. The term “ground truth” could thus be irritating, which is why we would consider the electrode coordinates only as reference points.
> To determine how well the learned EEG-specific variate embeddings reflect the true electrode layout of the 10-20 EEG system, we perform the following steps. Assume the 3D electrode coordinates of the 10-20 EEG system to be defined in Euclidean space $\mathbb{E}_Y^3$. We first project the EEG-specific variate embeddings into Euclidean space $\mathbb{E}_X^3$, then align them with the 3D electrode coordinates of the 10-20 EEG system in $\mathbb{E}_Y^3$ through multivariate linear regression, and eventually quantify their correlation by determining the coefficient of determination $R^2$.
>
> [13] Homan, R. et al. "Cerebral location of international 10–20 system electrode placement." Electroencephalography and clinical neurophysiology. 1987.

---

> ### Author Response · Authors · 2024-11-29
> **Author Responses (3/n)**
>
> (Q4) ***Is there any intuition behind what information could be shared across such diverse domains to make pre-training on them useful?***
>
> For OTiS, we employ a shared projection layer across all frequencies, variates, and domains, as we view this layer as a general feature extractor. We believe that low-level logic in time series, e.g. periodicity (or more simply, the pattern that a “low” is often followed by a “high”), can be learned across domains. We have analysed the time series of our diverse pre-training at the scale of a single patch (size 24) and found that, visually, they are indistinguishable at this scale: they all exhibit periodicity, regardless of the domain. We hypothesise that OTiS effectively captures such patterns across domains, which can then be leveraged during fine-tuning. This is particularly beneficial for domains with limited data, where the available data is often insufficient for learning such patterns with a randomly initialised model.

---

> ### Author Response · Authors · 2024-12-03
> **Final Author Responses**
>
> Dear Reviewer ***pz7i***,
>
> We would like to follow up on the discussion period to ensure that all of your points have been addressed by our rebuttal. In particular, we have worked to provide detailed responses to your questions regarding the (1) training time during fine-tuning, (2) downstream performance of models trained exclusively on the target data, (3) principal component analysis of the EEG-specific variate embeddings, and the (4) intuition behind the information shared across domains that makes pre-training on them beneficial.
>
> We hope our rebuttal, along with the discussion involving Reviewers ***UJog*** and ***tXLU***, has adequately addressed the points you raised. If you believe this to be the case, we would greatly appreciate a final adjustment of your scores to reflect this.
>
> Thanks again for your evaluation of our work and your valuable feedback.
>
> Authors

---

### Official Review · Reviewer_Aff6 · 2024-11-03

**Soundness:** 3
**Presentation:** 4
**Contribution:** 3
**Rating:** 6
**Confidence:** 4

**Summary:**

This paper presents OTiS, a pre-trained foundation model on large-scale time series data to support multi-tasks across domains. Extensive experiments are conducted to demonstrate the powerful performance of the foundation model. This paper is prepared in high quality and can be accepted.

**Strengths:**

1. This paper targets a very important research problem, the time series foundation model. Because the time series data has very high variance across different domains and different tasks. How to integrate them and train one foundation model remains challenging.
2. This paper has a very high quality of preparation. The writing, the organization, and the figure are prepared nicely and with enough details.
3. The results shown in Tab 1, 2, and 3 are competitive compared to baseline TS models.

**Weaknesses:**

1. Add a subsection to show which category baselines will be compared. For example, traditional TS model, deep learning, TS foundation model, etc.
2. I expect a comparison with some SOTA TS foundation models. For example, https://arxiv.org/abs/2405.02358 . If this part is added, that would be great.
3. Currently, the author use fine-tune to adapt the pre-trained model to various downstream tasks. Can you also add one more subsection to test the prompting on this TS foundation model? That would be another great point.

**Questions:**

See details in weakness.

---

> ### Author Response · Authors · 2024-11-29
> **Author Responses (1/n)**
>
> Thank you for your constructive comments and thorough evaluation. We hope the following clarifications and additional experiments adequately address the points raised.
>
> ---
> (1) ***Clarification of the baseline models***
>
> We have added a subsection to the experiments section, discussing the categories of the baselines. Moreover, we have included a summary of all baselines, detailing their architectures, pre-training strategies, and domain adaptation techniques, in Appendix B.
>
> ---
> (2) ***Comparison with state-of-the-art foundation models for time series analysis***
>
> We have compared our approach against six state-of-the-art foundation models, including Time-LLM [4], GPT4TS [5], MOMENT [6], MOIRAI [7], BIOT [8], and LaBraM [9], across classification and forecasting tasks, as discussed in the experiments section and Appendix G.2.
>
> [4] Jin, M. et al. "Time-LLM: Time Series Forecasting by Reprogramming Large Language Models." International Conference on Learning Representations (ICLR). 2023.
>
> [5] Zhou, T. et al. "One fits all: Power general time series analysis by pretrained lm." Advances in Neural Information Processing Systems (NeurIPS). 2024.
>
> [6] Goswami, M. et al. "MOMENT: A Family of Open Time-series Foundation Models." International Conference on Machine Learning (ICML). 2024.
>
> [7] Woo, G. et al. "Unified Training of Universal Time Series Forecasting Transformers." International Conference on Machine Learning (ICML). 2024.
>
> [8] Yang, C. et al. “Biot: Biosignal transformer for cross-data learning in the wild.” Advances in Neural Information Processing Systems (NeurIPS). 2024.
>
> [9] Jiang, W. et al. “Large brain model for learning generic representations with tremendous EEG data in BCI.” International Conference on Learning Representations (ICLR). 2024.
>
> ---
> (3) ***Evaluation of the prompting (i.e. zero-shot performance)***
>
> We have conducted further experiments to investigate the quality of the time series features extracted by our frozen model. These include zero-shot experiments for classification, linear probing for regression, and minimal tuning (< 1k trainable parameters) for forecasting, as detailed in the experiments section and Appendix F.

---

> ### Author Response · Authors · 2024-12-03
> **Final Author Responses**
>
> Dear Reviewer ***Aff6***,
>
> Thank you once again for your positive feedback on our study.
>
> We would like to follow up on the discussion period to ensure that all your points have been addressed in our rebuttal. Specifically, we have (1) added a subsection to categorise all baseline models, (2) provided comparisons with state-of-the-art time series foundation models [45], including Time-LLM [4] and GPT4TS [5], and (3) included a new subsection to evaluate our model in zero-shot settings.
>
> We hope these updates to the manuscript have adequately addressed the points you raised. If you agree, we would greatly appreciate a final adjustment of your scores to further support our study.
>
> Authors
>
> ---
> [4] Jin, M. et al. "Time-LLM: Time Series Forecasting by Reprogramming Large Language Models." International Conference on Learning Representations (ICLR). 2023.
>
> [5] Zhou, T. et al. "One fits all: Power general time series analysis by pretrained lm." Advances in Neural Information Processing Systems (NeurIPS). 2024.
>
> [45] Ye, J. et al. "A Survey of Time Series Foundation Models: Generalizing Time Series Representation with Large Language Model." arXiv preprint arXiv:2405.02358. 2024.

---

### Author Response · Authors · 2024-11-29
**Revised Manuscript Upload**

We thank all reviewers for their constructive and insightful efforts in evaluating this work, we truly believe our work has improved as a result of your suggestions. We have uploaded revised files, with several modifications:
1. more experiments, including new baselines and zero-shot evaluations;
2. revised experiments section and Appendix, to provide details on the baselines, experimental setup, and results;
3. more ablation studies, regarding the composition of the dual masking strategy and pre-training strategies;
4. more visualisations and examples in the Appendix, including domain signature analysis and latent space analysis.

The above points are marked in red (in both main paper and Appendix).
Besides these points, we have also revised figure captions, formulations, and other minor points in the manuscript. We have addressed the reviewers’ comments on a point-by-point basis below.

Thank you again for the constructive efforts in the comments and reviews,

Authors

P.S.: We are a bit late to the party, but hope for an active and lively discussion with the reviewers.

---

### Author Response · Authors · 2024-12-01
**Revised Manuscript Feedback**

Dear Reviewers,

We know you might have other plans on a Sunday, however, we would greatly value your feedback on our rebuttal.

In response to your insightful comments and questions, we have conducted additional experiments and provided detailed analyses in our rebuttal. Although these experiments required a significant part of the discussion period, we believe they have substantially improved the manuscript and we would welcome any further comments or discussion. If you believe that we have adequately addressed the points you raised, we would greatly appreciate an appropriate adjustment to your scores to reflect this.

Thank you once again for your constructive comments, which have truly enhanced the quality of our study.

Authors

---

### Comment · Area_Chair_QYK2 · 2024-12-02

Dear reviewers,

Could you please help to take a look at the author responses and let the authors know if your concerns have been addressed or not? Thank you very much!

Best regards,

AC

---

### Author Response · Authors · 2024-12-02
**Revised Manuscript Final Feedback**

Dear Reviewers,

We know you have already put a lot of time and effort into evaluating multiple manuscripts, including our study. As the journey is almost over, we would like to take this final opportunity to ask whether you have enjoyed:

Reviewer ***Aff6***: The detailed summary of the baselines, the comparison with SOTA foundation models, or the additional zero-shot evaluations?

Reviewer ***pz7i***: The more careful explanations of the results, the clarification of the principal component analysis, or our extensive comparison with domain-specific models?

Reviwer ***UJog***: The additional experiments on generalisabilty across domains, the analysis of Weather-specific variate emebddings, or the open discussion on the definition of domains?

If so, we would greatly appreciate a final adjustment of your scores to reflect this. Of course, we are happy to discuss any further comments or concerns, just as we have been engaging with Reviewer ***yUJh*** and Reviewer ***tXLU***.

Thanks once again for your constructive feedback and contributions, which have truly improved the quality of this study.

Authors

---

### Meta-Review · Area_Chair_QYK2 · 2024-12-23

**Metareview:**

This paper proposes OTiS, a pre-trained foundation model for multi-domain time series analysis, designed to handle the heterogeneity of variables and temporal dynamics across domains. The key contributions include a domain-specific tokenizer, a dual-masking strategy, and a novel loss function (NCC). Experimental results are presented across multiple tasks, including classification, regression, and forecasting, and the authors provide visualizations to highlight the interpretability of the learned embeddings.

This paper is working on a very challenging task – cross-domain time series analysis. The novelty of this work is good. The manuscript is generally well-written, and the proposed methodology is easy to follow. After the rebuttal, the evaluation part is also enhanced. However, the major concern after the rebuttal is still the experiment part. For example, reviewers pointed out that the experiments lack the comparison with other key methods such as UniTS. Experimental results are presented without sufficient explanations or analysis, and important experimental details are missing. In addition, multiple reviewers have concerns about the model design, e.g., the shared patch size may not be able to capture domain-specific temporal dynamics. For these limitations, I am inclined to recommend rejecting this paper.

**Additional Comments On Reviewer Discussion:**

During the rebuttal, 3 out 5 reviewers responded to the authors’ replies. Reviewer UJog increased the score to 5 as most of the concerns have been addressed during the rebuttal. Reviewer yUJh kept the original score as the author responses did not well address the concerns about the experimental results. Reviewer tXLU also kept the score due to the concerns about the experimental results. Overall, I agree with the reviewers and share the same concerns about the experimental part and thus I would like to recommend rejecting the paper.

---

### Decision · Program_Chairs · 2025-01-22

Reject